# RNAmetasome network for macromolecule biogenesis in human cells

Shiro Iuchi [1✉] & Joao A. Paulo [1]

RNA plays a central role in macromolecule biogenesis for various pathways, such as gene expression, ribosome biogenesis, and chromatin remodeling. However, RNA must be converted from its nascent to functional forms for that role. Here, we describe a large RNA metabolic network (RNAmetasome network) for macromolecule biogenesis in human cells. In HEK293T, the network consists of proteins responsible for gene expression, splicing, ribosome biogenesis, chromatin remodeling, and cell cycle. Reciprocal immunoprecipitations show that MKI67, GNL2, MDN1, and ELMSAN1 are core proteins of the network, and knockdown of either *MKI67* or *GNL2* affects the state of the other protein, MDN1, and some other network members. Furthermore, *GNL2* knockdown retards cell proliferation. Several proteins of the RNAmetasome network are diminished in Hela.cl1, and this diminishment is associated with low expression of MDN1 and elevated MKI67 degradation. These results together suggest that the RNAmetasome network is present in human cells and associated with proliferation, and that MKI67, GNL2, and MDN1 play an important role in organizing the RNAmetasome network.

[1] Department of Cell Biology, Harvard Medical School, 240 Longwood Avenue, Boston, MA 20115, USA. ✉email: shiro_iuchi@hms.harvard.edu

Crick defined central dogma of macromolecule biogenesis over 6 decades ago as genetic information transfer from DNA to DNA, DNA to RNA, and RNA to protein[1]. Yet, although the reverse transfer, i.e., from RNA to DNA or RNA, can take place in cells infected by RNA viruses, transfer from protein to RNA or DNA in all cells was not accepted. As Crick intended, central dogma has guided molecular biology and biologists to evade the chaos that could have happened at the dawn of the science. Crick deduced central dogma from the limited evidence available at that time. Since then, a great number of investigations have produced numerous findings on macromolecule biogenesis. These investigations have established that RNAs play various important roles in biogenesis pathways as messengers of DNA, material of ribosome, and regulators. These studies further found that most nascent RNAs, if not all, lack the capability of executing innate roles which are obtained after being processed by protein complexes[2–7]. As such, RNA metabolic processes are interwound and complicated. However, RNAs involved in these processes can be categorized into two groups based on their function: RNAs for macromolecule biogenesis pathways and RNAs for regulation of genes involved in these pathways. Topics belonging to the latter group are actively investigated at present, and the emerging results are inextricably linked to chromatin remodeling that is established and maintained by histone codes and DNA methylation/acetylation. Histone code is written by methylation, acetylation, and ubiquitylation and is responsible for the recruitment of activator or silencer proteins to specific DNA loci[8–12]. Among many proteins, histone deacetylases (HDAC1/2) and its associated proteins, ELM2 and SANT domain-containing protein 1 (ELMSAN1) and deoxynucleotidyltransferase terminal interacting protein 1 (DNTTIP1), were discovered to be mitotic deacetylases that may be responsible for regulation of gene expression[13]. The structure of the complex was then found to form a heterooctamer with ELM2-SAN domain positioned at $^{721}AA^{879}$ of canonical ELMSAN1 (1045aa, Q6PJG2)[14], and the complex binds to the nucleosome. Recently, the ELMSAN1/DNTTIP1/HDAC1 complex was found to play an important regulatory role in mouse neural differentiation by not only silencing expression of a group of genes via deacetylation of H3K27ac but also activating expression of another group of genes via deacetylation of H4K20ac[12]. During our continuous investigation of KDM2A, a H3K36me2 demethylase[15,16], we found that its DNA binding domain cluster counteracts binding of ELMSAN1 to a GC-rich DNA. During further investigation of its inhibitory mechanism, we encountered an inconsistency with other groups[12–14] on the subunits of the ELMSAN1 complex. The subsequent pursuit of the ELMSAN1 subunits led us to find an RNA metabolic network for macromolecule biogenesis consisting of hundreds of proteins. Accordingly, we have named it the RNAmetasome network. In this report, we describe the discovery of the RNAmetasome network and the characterization of its core proteins, proliferation marker protein Ki67 (MKI67), nucleolar GTP-binding protein 2 (GNL2), and midasin (MDN1).

## Results

**ELMSAN1 does not appreciably bind HDACs and histones.** The CXXC–PHD–NLSR (CPN) domain cluster of KDM2A binds the CpG-rich *Insulin Like Growth Factor Binding Protein Like 1* (*IGFBPL1*) gene promoter sequence and consequently stimulates binding of various human nuclear proteins to the sequence[16]. On the other hand, it also inhibits binding of many other proteins to the sequence. In that experiment, we prepared nuclear extracts from HEK293T cells expressing EGFP–CPN or EGFP and mixed each extract with the *IGFBPL1* promoter sequence immobilized to magnetic beads. Then, bound proteins were eluted and separated via SDS-PAGE. Finally, slices of the gel were excised and subjected to mass spectrometry analysis (MS). For the current interest as to what proteins are strongly inhibited by CPN, we picked the data from gel slices, 1–4 (Fig. 5a)[16], calculated the severity of the inhibition, and then plotted the resulting values on an X–Y diagram by combining affinity data of the proteins for the bait (Fig. 1a, Supplementary Data 1). This diagram enabled us to classify the inhibited proteins into two groups: proteins enriched less than 16-fold and those more than that. The former group of proteins was at least partly inhibited nonspecifically by CPN as a gray curve drawn in the upper part of the diagram indicates, and therefore, these proteins were not of interest to us. On the other hand, the latter group proteins appeared to be inhibited by a specific interaction between each protein and CPN on the bait, and therefore, these proteins were good candidates for further exploration. Of those, about 20 proteins were severely inhibited by CPN, and we found within this group an interesting protein (Fig. 1a, red dot), called ELMSAN1, which has a potential to regulate expression of the *IGFBPL1* gene.

ELMSAN1 forms a complex with DNTTIP1 and HDAC1/2 at a 1:1:1 molar ratio[13,14]. It also binds chromatin and regulates gene expression[12]. Accordingly, it was probable that KDM2A removes the ELMSAN1/DNTTIP1/HDAC1/2 complex to inactivate the *IGFBPL1* gene. To shed light on the regulatory mechanism, we attempted to recover the ELMSAN1/DNTTIP1/HDAC1/2 complex by immunoprecipitation (IP) with a rabbit ELMSAN1 antibody (Supplementary Table 1). This IP yielded 140-kDa (ELMS), 60-kDa (#8), and some other minor protein bands (Fig. 1b), along with DNTTIP1 (Fig. 1c), but unexpectedly it barely recovered the HDAC1, HDAC2, H3, and H2B. This result could be valid or a false positive result associated with the antibody. Accordingly, we evaluated quality of the antibody and confirmed that this antibody is highly sensitive and specific to ELMSAN1. The confirmation was performed as follows: First, three different siRNAs prepared for knockdown of the canonical ELMSAN1 transcript (Supplementary Table 2) diminished 140-kDa protein (Fig. 1d), suggesting that the 140-kDa protein is the canonical ELMSAN1 (Q6PJG2). Second, epELM-5 (Supplementary Fig. 1a, b), i.e., a 6x His-tagged epitope ($^{550}AA^{600}$ of Q6PJG2) bound to a piece of PVDF, inhibited the antibody from recognizing the 140-kDa ELMSAN1 and 60-kDa #8 of HEK293T (arrowhead of Fig. 1e). Likewise, epELM-5 inhibited the antibody's recognition of other bands. This inhibition looked weaker than expected on a few bands, but this weakness is caused by both binding of epELM-5 to the bands and absorption of the epitope to the background. These diminished bands are likely some of the five known ELMSAN1 isomers (the UniProt database) and their partially degraded fragments. Unlike these proteins, band #2 was not affected by epELM-5. An equivalent band of another cell line, Hela.cl1, was also not affected. This result may lead one to presume that the antibody cross recognized #2. However, #2 is likely an isomer of ELMAN1 with very high affinity for the antibody that could be caused by a posttranslational modification like the event occurring to mucin1[17]. This interpretation is consistent with two facts: (1) the epitope ($^{550}AA^{600}$ of canonical ELMSAN1) is unique among human proteins as blastp search against the NCBI human protein database hits no other peptide sequence, and (2) the antibody is affinity purified. Third, the immunoprecipitated proteins were also recognized although weakly with a mouse monoclonal ELMSAN1 antibody, except for #8 (Fig. 1f). #8 had another peculiarity. It had an inverse relationship with the 140-kDa ELMSAN1 between the HEk293T and the Hela.cl1 (Supplementary Fig. 1c), suggesting that #8 is an ELMSAN1 isomer, H7C1L3, that has most of $^{550}AA^{600}$ but not $^{631}AA^{657}$. Fourth, Hela.cl1

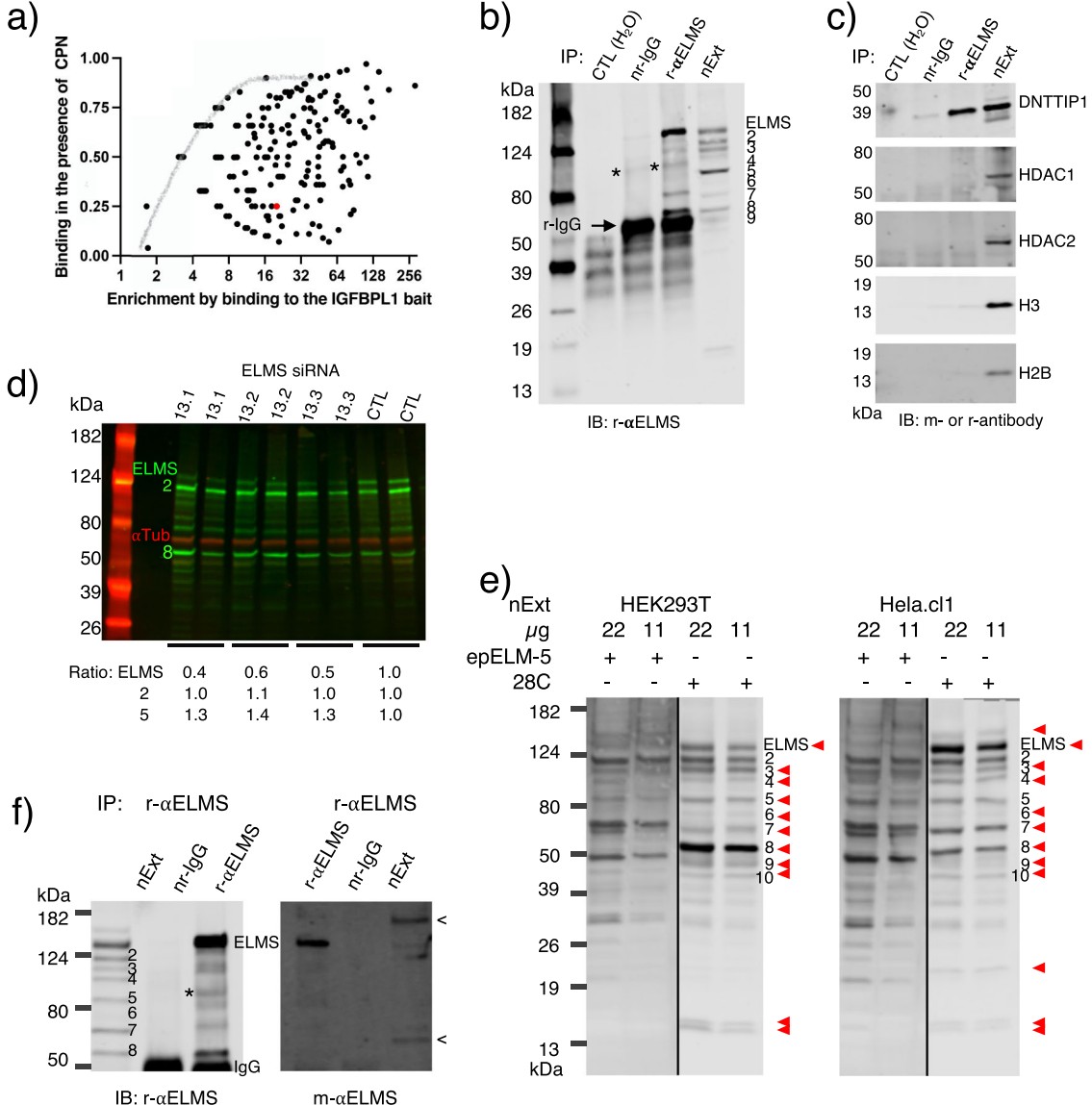

**Fig. 1 Immunoprecipitation (IP) with the rabbit anti-ELMSAN1 antibody. a** ELMSAN1's binding to the IGFBPL1 bait is strongly inhibited by the KDM2A CPN domain. ELMSAN1 is shown by a red dot. Values on **a** are calculated from data of the previous publication: (gel slices #1–4 of Fig. 5a; -CPN nuclear extract of 5c)[16]. $Y$ axis shows residual binding level of proteins in the presence of the CPN domain by ratio ($C_{b+CPN}/C_{b-CPN}$), and $X$ axis shows enrichment of proteins by fold [$(C/SOC)_{b-CPN}/(C/SOC)_{e-CPN}$] with the nuclear extract lacking the CPN domain. $(C/SOC)_{b-CPN}$ and $(C/SOC)_{e-CPN}$ are concentration of a protein bound to the bait and the protein in the nuclear extract, respectively. Data for $Y < 1$ are plotted. **b, c** IP of nuclear extract with a rabbit anti-ELMSAN1 antibody. The IP is visualized by IB. nr-IgG (normal rabbit IgG), r-αELMS rabbit (anti-ELMSAN1 antibody), nExt (nuclear extract input). An asterisk (*) indicates noise derived from IgG. This symbol (*) was used throughout this report to indicate noise. The host of the antibodies used for the IB is either rabbit (r) or mouse (m). **d** Knockdown of the ELMSAN1 gene. Three different siRNAs and one universal negative control were added to 10 nM in duplicate cultures, respectively. Efficacy of the knockdown is estimated by dividing the level of ELMSAN1 with that of α-tubulin. **e** Inhibition of the rabbit ELMSAN1 antibody by the 6x His-tagged epitope (epELM-5). Twenty micrograms of epELM-5 bound to PVDF, and the control (28C) bound to PVDF (see Supplementary Fig. 1) were incubated in the IB solution that has both the r-anti-ELMSAN1 antibody and a sample PVDF membrane. The red arrowhead indicates the band for which binding of the antibody was reduced. **f** Identification of ELMSAN1 with a mouse monoclonal anti-ELMSAN1 antibody. PAGE separated immunoprecipitates were transferred to PVDF and probed with the rabbit (left) and the mouse monoclonal anti-ELMSAN1 antibody (right). Arrow (<) shows unidentified bands detected with the mouse antibody in the nuclear extract.

expressed a ELMSAN1 band that migrated more slowly than 140-kDa ELMSAN1 (Supplementary Fig. 1e). This band is likely another isomer, A0A1C7CYX1, with a 1099 long amino acid sequence. Taken together, these results demonstrate that the rabbit anti-ELMSAN1 antibody captures the canonical ELMSAN1 (Q6PJG2), a few isomers, and their fragments but does not recognize foreign proteins. Thus, this antibody is suitable for IP and western blotting of ELMSAN1. We disregarded a few bands that were recognized with the antibody only under the presence of

epELM-5, as these were likely foreign proteins bound by the epitope liberated from the PVDF during the probing.

**ELMSAN1 belongs to a macromolecule biogenesis network involving hundreds of RNA metabolic proteins.** As ELMSAN1 did not appreciably co-immunoprecipitate HDAC1/2, we wondered what proteins ELMSAN1 interacts with and then searched for ELMSAN1-interacting proteins by MS of the immunoprecipitated

protein complex. The immunoprecipitates contained 461 twofold or greater enriched proteins (Supplementary Fig. 2a, Supplementary Data 2). Another co-IP with the ELMSAN1 antibody resulted in enrichment of 582 proteins (Supplementary Fig. 2b, Supplementary Data 2), of which 262 were reproducible. Using these data, we next made a protein–protein interaction network for each set by independently importing proteins from the STRING database with confidence cutoff 0.999 via Cytoscape. This exploration found that the two resulting networks are similar and complementary to each other. Accordingly, we combined the two datasets and obtained a more comprehensive network (Fig. 2). It consisted of ten functional groups (modules): (1) transcription factors, (2) splicing factors, (3) CCR4-NOT transcription complex (CNOT complex), (4) decapping proteins, (5) RNA exosome complex, (6) small ribosomal subunit assembly proteins, (7) ribosome proteins with EIFs, (8) large ribosomal subunit assembly proteins (rixosome), (9) GTP-binding protein-rich group, and (10) cell cycle proteins. This result revealed, with a 0.001 false discovery rate (FDR), that hundreds of different classes of proteins responsible for RNA metabolism form the macromolecule biogenesis network. Therefore, we named this network the RNAmetasome network. In addition to these proteins, some protein groups that were not assigned in this RNAmetasome network were present in the enriched protein complex. They include replication factor C subunits (RFCs), AA-tRNA ligases, general transcription factor 3Cs, chromatin remodeling factors (SMARCAs, SMARCD1, and BAZ1s), and siRNA producing protein group (Supplementary Fig. 3). We include these complexes as members of the RNAmetasome network. Curiously, the most extensively enriched proteins, such as MKI67 and MDN1, were not assigned to the main network (Fig. 2, #11). We analyzed these two datasets, consolidating the datasets using SAINT[18]. The output of the analysis showed that 1417 proteins had an empirical fold change score (FC-A) > 1, essentially equivalent to SAINT probability score ≥ 0.5, (Supplementary Data 3), and a network drawn for these proteins by Cytoscape-STRING completely covered the RNAmetasome network of Fig. 2 (Supplementary Fig. 4) and further added more functional groups, some of which were previously isolated subgroups (Supplementary Fig. 3) of the network. Notable change associated with the result by SAINT was the integration of MDN1 and gene silencing group proteins (HDAC2 and PRC, etc.) into the RNAmetasome network. During of a series of experiments, we noticed that many of co-immunoprecipitated proteins were precipitated also with a normal rabbit IgG (nr-IgG), with some proteins exceeding the level of the proteins precipitated with the anti-ELMSAN1 antibody. To determine whether this unusual IP result is a false positive, we selected eight diverse proteins including three proteins that are apparently more highly enriched with the normal IgG (NAT10, MKI67, ELMSAN1, MDN1, YTHDC2, NSD2, GNL2, and TOP2B) and performed IP–IB (Supplementary Fig. 2d). The result demonstrated that these proteins were indeed precipitated with the nr-IgG; however, importantly, all examined proteins were precipitated to a lesser extent with the nr-IgG than the anti-ELMSAN1 antibody. Thus, our conclusion concerning the RNAmetasome network remains valid. Following these analyses, we performed another duplicate IP–MS experiment (Exp #3 and #4), having two kinds of controls for each experiment: water and nr-IgG in place of the rabbit ELMSAN1 antibody. This approach would remove both overestimated and underestimated proteins. Analysis of the result by SAINT increased the number of FC-A > 1 proteins to 1700 due to better recovery of proteins (Supplementary Data 4). The network drawn by Cytoscape-STRING with these proteins not only reconfirmed that the RNAmetasome network is present in HEK293T but also revealed that new functional subgroups are involved in the network (Supplementary Fig. 5). Examples are protein groups for degradation of mRNA (CPSF complex, #12), export of mRNA and polyadenylated and spliced RNA (THOC1

complex, #13), protein degradation (PSMC complex, #14), and DNA replication (MCM and RFC, #15). In addition, chromatin remodeling complexes (SMARCs, HDAC2, CHD3, and a few PRC proteins, #16) and siRNA producing protein groups (DICER1, AGOs, and DROSHA, #17) were found to form a modestly sized network, separated from the main RNAmetasome network. Finding a few PRC proteins (SUZ12, EED, and JARID2) in this chromatin remodeling complex module suggests that the cutoff of FC-A > 1 was suitable for identifying RNAmetasome network constituent proteins, as recent research focussing on rixosome proteins found that rixosome proteins interact with PRC members to silence the expression of a certain genes[7].

Next, we examined what methyltransferases participate in the RNAmetasome network. Histone methyltransferases are of great interest as they are principal measures for the regulation of gene expression and chromatin remodeling for differentiation and development[3,10,11,19]. Of the 60 known protein methyltransferases[20], only 4 (NSD1, NSD2 responsible for methylation of H3K36[21,22], KMT2A responsible for methylation of H3K4, and EZH2 responsible for methylation of H3K27) were found in the ELMSAN1 immunoprecipitates. On the other hand, five out of six KMT2 isomers[23] and SUV39H1/2 were not included (Supplementary Data 2–4). Meanwhile, EZH2, SUZ12, and JARID2 were present at the very low level in the immunoprecipitates, and other PRC subunits were not found. Fifteen other types of methyltransferases and their accessory proteins were present in the RNAmetasome network. Of those, 12 were methyltransferases of rRNA and tRNA, and 2 were accessory proteins. In addition, the RNAmetasome network contained other RNA modification proteins, such as members of snoRNPs containing NOP56/NOP58/NHP2/snoRNAs complexes, cytidine acetyltransferase, NAT10, and polynucleotide 5′-kinase, NOL9, and $N^6$ methyladenine binding proteins, YTHDC2 and YTHDF1/2/3[24]. However, it contained none of three components, METTL3, YTHDC1, and SETB1, that are responsible for heterochromatin formation to silence endogenous retrovirus genes via its $N^6$ methyladenine of RNA[25]. Thus, our results again point to the conclusion that the RNAmetasome network focuses on implementing rRNA, tRNA, mRNA metabolism, its downstream metabolism, and their regulation for macromolecule biogenesis.

Note that NSD2, which transfers two methyl groups to the K36 residue of H3[21,22], was consistently found in the immunoprecipitates, suggesting that NSD2 is an important member of the RNAmetasome network. As the KDM2A CPN domain excludes ELMSAN1 from the *IGFBPL1* promoter, the domain is likely to eliminate NDS2 from the promoter as well. Therefore, the level of H3K36me2 at CpG-rich regions is controlled by competition between NSD2 and KDM2A with respect to: (1) the enzyme activity between the methyltransferase and the demethylase reaction, and (2) binding of the two enzymes to the CpG-rich promoter sequences.

**MKI67 is a scaffold of RNAmetasome network.** Unexpectedly, this RNAmetasome network (Fig. 2) excludes four of the most extensively enriched proteins, MDN1, MKI67, DNJC13, and YTHDC2 (Fig. 3a, Exp #2), whereas it included the fifth top protein, GNL2 (Fig. 2, subgroup #9, green). No accommodation of MDN1 was especially puzzling as MDN1 binds PELP1 (Rix1) of the rixosome complex and is responsible for the maturation of the 60S ribosome by its ATP-dependent conformation change[5,7,26–28]. This could mean that these four proteins were enriched by cross reactivity of the rabbit ELMSAN1 antibody. To ensure our result was derived from co-IP but not a cross reaction, we performed reciprocal co-IPs of ELMSAN1 with a rabbit MDN1, a rabbit MKI67, and a rabbit YTHDC2 antibody, using aliquots of a pooled nuclear extract. As expected, these three

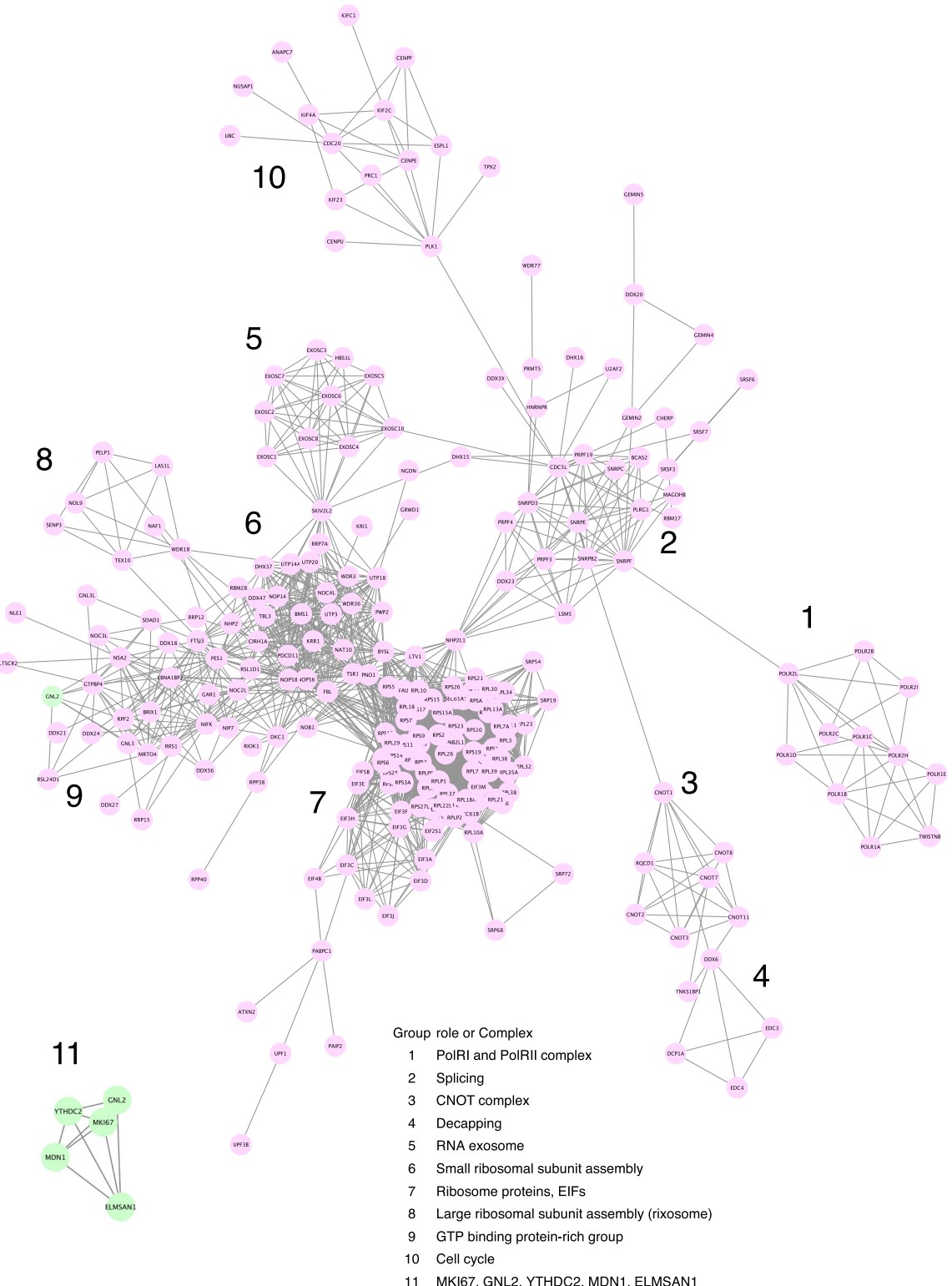

| Group | role or Complex |
|---|---|
| 1 | PolRI and PolRII complex |
| 2 | Splicing |
| 3 | CNOT complex |
| 4 | Decapping |
| 5 | RNA exosome |
| 6 | Small ribosomal subunit assembly |
| 7 | Ribosome proteins, EIFs |
| 8 | Large ribosomal subunit assembly (rixosome) |
| 9 | GTP binding protein-rich group |
| 10 | Cell cycle |
| 11 | MKI67, GNL2, YTHDC2, MDN1, ELMSAN1 |

**Fig. 2 RNAmetasome network.** This network contains continuously edged proteins only. Proteins shown in green (#11) are the most extensively enriched but not included in this network according to the STRING database. The exception is GNL2, which is integrated as a member of the network.

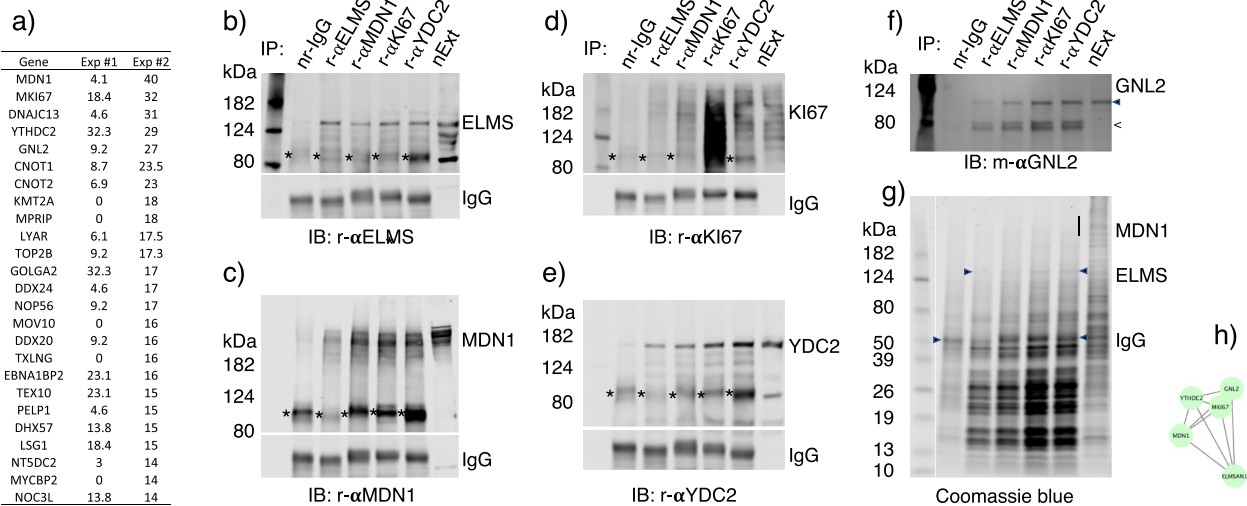

**Fig. 3 Reciprocal IP. a** Enrichment of proteins by two independent IP–MS experiments. The top 25 highest enriched proteins in Exp #2 and Exp #1 are shown in the table. **b–f** IP was carried out with rabbit anti-ELMSAN1 (r-αELMS), rabbit ani-MDN1 (r-αMDN1), rabbit anti-MKI67 (r-αKI67), and rabbit anti-YTHDC2 (r-αTDC2) antibodies, respectively. IB of the immunoprecipitates was carried out with these antibodies and mouse monoclonal anti-GNL2 (m-αGNL2) antibody, respectively. The bands indicated by < appear to have occurred during the experimental process. **g** Coomassie blue staining of the immunoprecipitates. **h** A model of the complex consisting of MKI67, GNL2, YTHDC2, MDN1, and ELMSAN1. This model is derived from evaluation of the reciprocal IP yields (Supplementary Table 3).

antibodies coprecipitated ELMSAN1(Fig. 3b). Likewise, all four antibodies, including the ELMSAN1 antibody, mutually coprecipitated three nondirect target proteins and GNL2 (Fig. 3c–f). Furthermore, immunoprecipitates captured with all four antibodies exhibited essentially the same profiles on SDS-PAGE (Fig. 3g). Thus, five of these proteins (ELMSAN1, MDN1, MKI67, YTCHDC2, and GNL2) interact with each other, and tether many other proteins. A quantitative analysis of this co-IP result (Supplementary Table 3) suggests that MKI67 is a scaffold protein to which the four proteins bind with proximity order of GNL2 > YTHDC2 > MDN1 > ELMSAN1 (Fig. 3h; also shown in Fig 2, #11). As GNL2 (Fig. 2, green) is highly edged with three modules (Fig. 2 subgroups #6, 8, 9), we predict that MKI67, YTHDC2, and MDN1(Fig. 2, subgroup #11) together with GNL2 play an important role in ribosome biogenesis. We disregarded DNAJC13 for this analysis as it localizes principally in the nuclear periphery and cytoplasm.

**MKI67 and GNL2 are involved in pre-60S ribosome maturation and chromatin organization.** As MKI67 is the scaffold protein of the RNAmetasome network, we sought to obtain evidence that MKI67 plays an important role in the RNAmetasome network. To this end, we took a cytogenetic approach. First, we double stained HEK293T cultures with the rabbit MKI67 antibody (epitope, KKAEDNVC at $^{3234}AA^{3241}$) and a mouse monoclonal antibody or with the mouse monoclonal MKI67 (epitope, 8x FKEL at $^{1105}AA^{2562}$)[29] and a rabbit antibody. Curiously, MKI67 stained with the rabbit antibody was observed as nm-order size sphere bodies (SBs), but it was also observed as a larger entity, specifically, as a line at the nucleolar AT-rich chromatin periphery (Fig. 4, panels 2, 10, 26). This observation suggests that MKI67 is present basically as small SBs, and its large population forms a large entity. This protein is also recognized by the mouse monoclonal MKI67 antibody although its stain was weak (Fig. 4, P18, 34). The weak stain can be rationalized by its weak affinity for its epitope on MKI67 and by the epitope's location in the protein molecule. *MKI67* gene knockdown diminished the nuclear MKI67 stained with either antibody (Fig. 4, P6, 14, 22, 30, 38). As such, both antibodies faithfully

reveal the presence and location of MKI67. Using NIFK that is known to bind MKI67 at its phosphorylated FHA domain[30], we investigated its colocalization with MKI67 as a positive control. NIKF colocalized with MKI67 at the nucleolar chromatin periphery, whilst it did not necessarily colocalize in the nucleoplasm (Fig. 4, P1–4), suggesting that the colocalization of the two proteins is conditional. Keeping this result in mind, we investigated the colocalization of other RNAmetasome network proteins with MKI67. GNL2 (P9–12), MDN1 (P17–20), DICER1 (P25–28), and YTHDC2 (P33–36) colocalized with MKI67 in the nucleolus, nucleoplasm, and/or nuclear periphery. ELMSAN1 and DNTTIP colocalized with MKI67 predominantly on and near the nuclear periphery where heterochromatin is associated with the inner nuclear membrane[8], but these were certainly present in the nucleoplasm as well (Supplementary Fig. 6). Among these proteins, GNL2 most extensively colocalized with MKI67 everywhere in the nucleus. All these proteins were recognized in small SBs like MKI67 in the nucleus, except for YTHDC2 that was recognized in noticeably larger SBs.

Next, we deciphered whether or not the colocalization reflects a protein–protein interaction by protein depletion. We expected that some RNAmetasome network members are dysregulated upon depletion of MKI67 as depletion of either subunit of the XRCC5/6 complex destabilizes the other subunit[31]. Knockdown of the MKI67 gene efficiently depleted MKI67 from almost every nucleus with the siRNAs (Supplementary Fig. 7, P4). In these nuclei, NIFK, GNL2, and DICER1 were diminished (Fig. 4, P5–8, 13–16, 29–32), while MDN1 behaved differently from the three proteins. MDN1 was diminished in the nucleolus and congregated in the nucleoplasm (Fig. 4, P21–24). These results suggest that NIFK, GNL2, MDN1, and DICER1 physically interact with MKI67 in the nucleus and achieve certain biological functions together. We failed to find any changes in the level and the localization of YTHDC2, but it does not indicate that YTHDC2 does not interact with MKI67.

To analyze further the significance of these protein–protein interactions, we depleted GNL2 with the siRNAs (Supplementary Fig. 7, P8). Then, with the rabbit antibody, we found that MKI67 occupies the periphery of the enlarged nucleolus while it is diminished elsewhere in the nucleus (Fig. 4, P53–56,

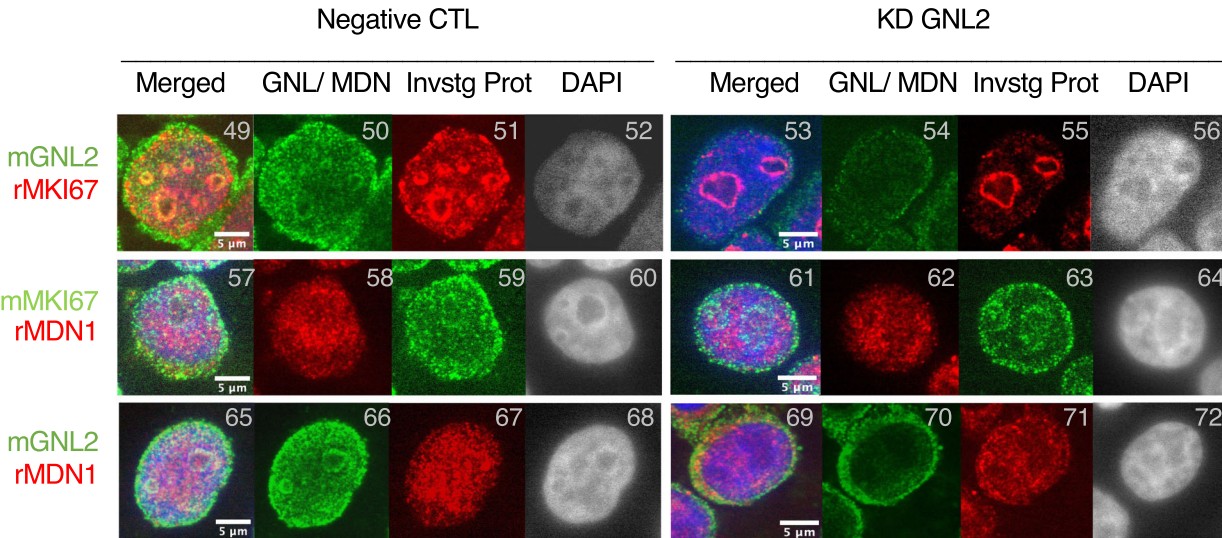

**Fig. 4 Double immunostaining of HEK293T cultures.** Gene knockdown was performed with 10 nM universal negative control siRNA (left,) and 10 nM siRNA for *MKI67* and *GNL2* (right). Red and green show proteins stained with rabbit antibodies and mouse antibodies, respectively. Rabbit and mouse antibodies used for probing are shown to the far left in red and green. Invstg Prot column shows immunostaining of primary target proteins for the investigation.

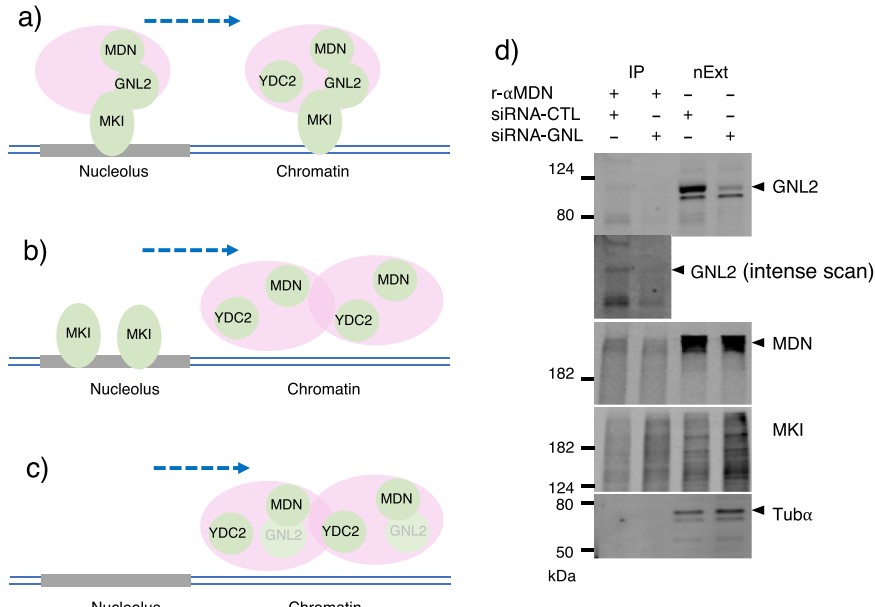

**Fig. 5 Migration of nucleolar proteins, MKI67, MDN1, and GNL2. a** A model of RNAmetasome network protein migration in HEK293T. Nucleolar proteins with MDN1 are maintained in the nucleolus by binding of MKI67 to the nucleolus periphery chromatin. The MDN1 complex migrates out of the nucleolus and recruits some proteins in the nucleoplasm, such as YTHDC2. **b** Depletion of GNL2 relaxes the interaction between MKI67 and MDN1 and accelerates migration of the MDN1 complex to the nucleoplasm. **c** Depletion of MKI67 accelerates migration of the MDN1 complex to the nucleoplasm. **d** Co-IP of GNL2 and MKI67 with the anti-MDN1 antibody in GNL2 depleted nuclear and the control extracts. The siRNA and the universal negative control siRNA were added at 40 nM.

Supplementary Fig. 7, P7). With the mouse antibody, inside of the dysregulated MKI67 appeared to be also occupied with the protein (Fig. 4, P61–64). In these nuclei, a majority of MDN1 was localized apart from MKI67 (Fig. 4, P61–64). Likewise, the peculiar localization of MDN1 was observed in DAPI stained nuclei (P69–72). These results suggest that MKI67 requires GNL2 for its migration from the nucleolus toward the nucleus, while MDN1 migrates independently of both GNL2 and MKI67 from the nucleolus (Fig. 5a, b). As MKI67 depletion diminishes the MDN1 nucleolar localization and deposit MDN1 to the nucleoplasm (Figs. 4, P21-24, and 5c), MKI67 and GNL2 together appear to hold MDN1 to prevent MDN1 from prematurely migrating out of the nucleolus or overproducing mature ribosomes. To evaluate this model, we depleted GNL2 and performed an IP of MDN1. The depletion decreased GNL2 by 90% (Fig. 5d) and the IP of MDN1 did not precipitate GNL2, proving that GNL2-free MDN1 can migrate from the nucleolus to the nucleoplasm. On the other hand, MDN1 still co-immunoprecipitated MKI67 (Fig. 5d), suggesting that MKI67 and MDN1 can interact with each other via RNAmetasome network proteins regardless of the presence or absence of GNL2 unless both proteins are physically separated by any measures like by nucleolus. GNL2 depletion increased the number of AT-rich loci[32] in the nucleus (Supplementary Fig. 7, P9).

**RNAmetasome network in Hela.cl1.** We postulated as to whether the RNAmetasome network exists in different human cell lines or if it is specific to HEK293T. If this metabolic network was important in human cells, it would be present in other human cell types, too. Prompted by this idea, we carried out IP–MS using Hela.cl1 nuclear extract and found that the RNAmetasome network is also present in this cell type (Fig. 6a, Supplementary Data 5). However, many of the network members were precipitated to a lesser extent in the Hela.cl1 extract than the HEK293T extract. Examples include: GNL2, CNOT1, CNOT9, NOP2, KRI1, TOP3B, XRN2, NIFK, CHD6, CDC5L, FTST3,

SENP3, and AGO2 (Supplementary Data 5), most of which are involved in splicing and processing of rRNA and mRNA. Some of these proteins were estimated to be absent, for example, GNL2, by the IP-MS (Supplementary Data 5). Then, we realized that MDN1 is absent from both the nuclear extract and immuno-precipitates. We subsequently confirmed by IP–IB that MDN1 was lower by two thirds in the Hela.cl1 than the HEK293T extract (Fig. 6b), and that co-IP of MDN1 with the anti-ELMSAN1 antibody was low and recognized only by a high-intensity scanned image (Fig. 6e, f). An additional and unexpected result with the Hela.cl1 extract was that MKI67 was severely degraded (Fig. 6c), so that only intact and less degraded MKI67 were immunoprecipitated with the anti-ELMSAN1 antibody, suggesting the possibility that nonimmunoprecipitated MKI67 completes against intact and less degraded MKI67 for protein binding. These features of MDN1 and MKI67 explain why the IP–MS estimated no co-IP of GNL2 although this protein was present in this nuclear extract at levels similar to that of the HHEK293T extract (Fig. 6d). Consistent with the IP–MS result, IP–IB assessed that GNL2 was immunoprecipitated at a very low level (Fig. 6f). Thus, GNL2 participation in the RNAmetasome network is under control of MDN1 and MKI67. The same mechanism should be applied to the other RNAmetasome proteins that were immunoprecipitated to a lesser degree in Hela.cl1 extract. With these results, we derive two conclusions: first, the RNAmetasome network is present in human cells, and second, content of the RNAmetasome network may be determined by MDN1 and MKI67 together with GNL2.

**Discussion**

Besides differences in the level of RNAmetasome constituent proteins, HEK293T and Hela.cl1 have another difference, i.e., involvement of nonspecific binding. RNAmetasome network proteins were precipitated to some extent with the nr-IgG when HEK293T nuclear extract was used, whereas these proteins were not precipitated with the same IgG when Hela.cl1 nuclear extract was used (Fig. 6e, f,

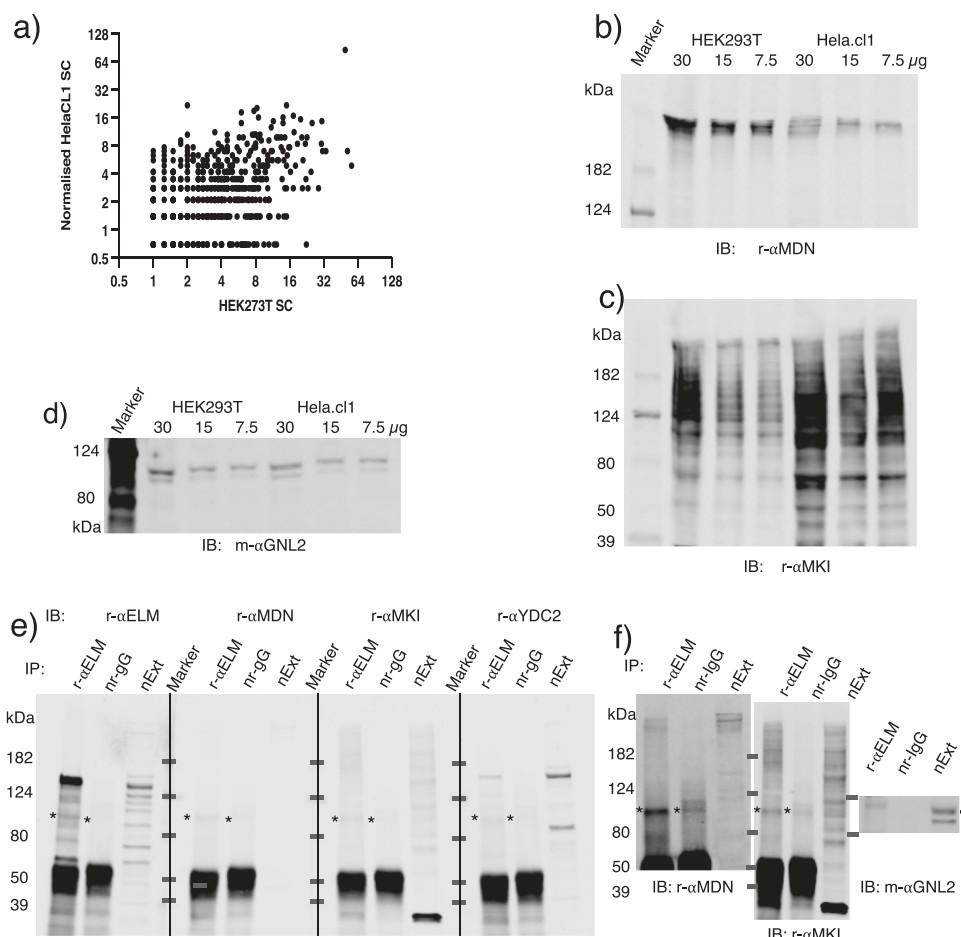

**Fig. 6 RNAmetasome network of Hela.cl1. a** Scatter diagram of the RNAmetasome network proteins. *X* and *Y* axes show net spectral count (SC) of the HEK293T and the Hela.cl1 proteins immunoprecipitated with the anti-ELMSAN1 antibody. Net SC of HEK293T = average of total SC − average of four control SC; likewise, net SC of Hela.cl1 = total SC − control SC. Net SC of HEK293T and Hela.cl1 were 14 and 10, respectively. In this diagram, 631 proteins are plotted after the Hela.cl1's values are normalized by factor 14/10. *r* = 0.47, *P* < 0.0001. **b–d** MDN1, MKI67, GNL2 level of HEK293T and Hela.cl1 nuclear extracts estimated by western blotting. **e** IB of proteins immunoprecipitated with the anti-ELMSAN1 antibody. All procedures for these experiments were carried out in parallel, which reveals that MDN1 is at an undetectable level. This ELMSAN1 IB image (far left) is flipped and presented in Fig. 1f (left). **f** High-intensity scanned image of **e** and GNL2.

Supplementary Fig. 8). As a normal IgG is traditionally used to search for specific binding, proteins mediated only by nonspecific binding should not be included as a factor for the network formation, and the portion mediated by nonspecific binding should not be considered as a part of protein–protein interaction. However, we questioned whether the traditional practice can be applied to our experimental results and rather tend to think that nonspecific binding of our observation contributes to the formation of the RNAmetasome network to some extent. Our nonspecific binding of a protein has a proportional relationship with specific binding of the protein to ELMSAN1 in general, i.e., the stronger the specific binding, the stronger the nonspecific binding (Supplementary Fig. 8). On the other hand, this relationship is not always applicable. For example, nonspecific binding of ELMSAN1 and some other proteins to the nr-IgG is negligible. Therefore, these results suggest that a protein with capability of specific binding to ELMSAN1 tends to be sticky, i.e., nonspecific binding-prone. Specific binding via domain-domain interaction plus nonspecific binding would make both protein–protein and complex–complex interactions stronger, and the latter would help specific binding revive when needed after its transient separation. We do not know how only the HEK293T nuclear extract facilitates the high nonspecific binding, but the nonspecific binding of HEK293T may be a measurable form of protein phase separation that has been recently recognized as an

important mechanism in forming droplets/membraneless organelles consisting of multiple proteins[33–35]. Consistent with this assumption, the RNAmetasome network proteins are present in the nucleus as SBs, a droplet-like structure, and our examination by PrDOS[36] on some RNAmetasome proteins predicts that these proteins have intrinsically disordered regions (IRDs), which is necessary for phase separation, along the molecule (Supplementary Fig. 9). Notably, MKI67 is one of the stickiest proteins of all RNAmetasome network proteins (Supplementary Fig. 2d, Supplementary Data 4) and has IRDs along the entire 358-kDa molecule. For these reasons, we believe that the nonspecific binding that we observed contributes to the formation of the RNAmetasome network to some extent. However, it is important to address that the RNAmetasome network is formed without the nonspecific binding.

One group reported a protein–protein interaction network consisting of 408 proteins[37], which shares similarity with the RNAmetasome network. Specifically, their complex includes 131 proteins that are present in the RNAmetasome network. However, their complex lacks YTHDC2, MDN1, ELMSAN1, and certain subgroups and does not contain sufficient proteins to form modules, such as CNOT, RNA exosome, and several cell cycle proteins. The difference between this research group and us would be derived from the choice of the direct target protein for the IP or from differences in cell line.

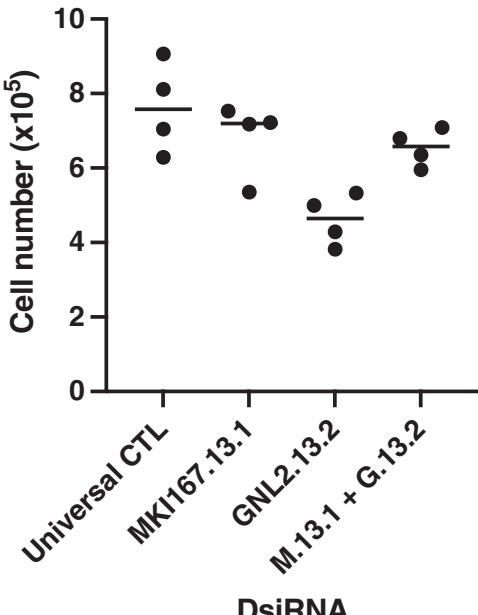

**Fig. 7 Effect of *MKI67* and *GNL2* knockdown on proliferation of HEK293T.**
Each siRNA was added at 20 nM in a culture with $1 \times 10^5$ cells, except for the double knockdown where each siRNA was added at 10 nM to make the total concentration 20 nM. Each dot on the diagram shows cell number of an independent culture. Results of the *t*-test for CTL KD vs MKI67 KD, CTL KD vs GNL2 KD, MKI67 KD vs GNL2 KD, and GNL2 KD vs MKI67 KD + GNL2 KD are $P = 0.34$, 0.005, 0.010, and 0.003, respectively.

The RNAmetasome network is assembled with hundreds of selected proteins. Among these proteins, MKI67, MDN1, and GNL2 play an important role in organizing the RNAmetasome network that would have an advantage over a protein-free diffusion system in coordinating a series of protein–protein interactions and enzymatic reactions. With this concept, we anticipated that knockdown of both *MKI67* and *GNL2* would have slowed cell proliferation. However, knockdown of *GNL2* retarded cell proliferation, whereas that of *MKI67* did not, and the double knockdown of the genes alleviated the *GNL2* knockdown retardation effect (Fig. 7). As such, the *MKI67* knockdown exhibited no obvious consequence in cell proliferation against our prediction, and this result on the *MKI67* knockdown is consistent with the observation that MKI67 null mutants grow as rapidly as the wild type[37]. Interestingly, however, we observed that knockdown of either the *MKI67* or the *GNL2* prompted MDN1 to leave the nucleolus for the nucleoplasm (Figs. 4, P21–24, P61–64, P69–72 and 5b, c), and knockdown of the *GNL2* retained MKI67 causing unusual chromatin distribution in the nucleus (Supplementary Fig. 7, P7–9). Therefore, we do not ignore the possibility that accumulation of MKI67 in the nucleolus may participate in controlling proliferation via chromatin arrangement. We are currently at the beginning of the RNAmetasome network research. Further genetic, biochemical, and cytological dissections are pivotal to gain insight into RNAmetasome network formation and macromolecule biogenesis control.

## Methods

**Cell culture.** HEK293T (ATCC CRL-3216) and Hela.cl1 cells were cultivated in DMEM (Thermo Fisher Scientific) supplemented with 10% fetal bovine serum (HyClone). Hela.cl1 was a gift cell line from Howard Green. The cells, the cultivation, and the handling procedures have been approved by Harvard University Committee on Microbiological Safety (ID: 20-155). We carried out experiments under the NIH rDNA and the Federal Occupational Safety and Health Administration Bloodborne Pathogen Standard Guidelines.

**Cloning of DNA encoding the epitope for the rabbit ELMSN1 antibody and preparation of the epitope.** We purified total RNA from HEK293T cultures, and synthesized cDNA in ProtoScript II mixture (NEB, E6560S), using d(T)23VN and 600 ng total RNA. The piece of DNA encoding epitope, $^{550}AA^{600}$ of ELMSAN1 (Q6PJG2), was amplified in a mixture with two primers, Olit351 and Olit352, the cDNA, and Q5 hot start high-fidelity DNA polymerase (NEB, M0494S), and then cloned into the BamHI-HindIII site of pET28C (+). Competent bacteria (NEB C3013I) were transformed with the cloned plasmids and the empty vector was and induced for production of the epitopes with 0.5 mM IPTG in LB for 4 h. The bacteria were collected by centrifugation, suspended in the extraction buffer with a complete protease inhibitor cocktail (Sigma), and sonicated. Protein concentration was estimated by BCA. Plasmid constructed in this work, pET-epELM-5, is available from Addgene, ID: 179391.

**Preparation of nuclear extracts.** Preconfluent HEK293T or Hela.cl1 culture in a 100 mm dish was washed with 10-ml cold PBS. The culture was scraped following addition of 1 ml cold PBS with 1x Complete EDTA-free protease inhibitor cocktail (Roche). Such cell suspensions prepared from a total of 12 dishes were mixed, and distributed to $12 \times 1.5$ ml tubes and centrifuged at 4000 rpm for 5 min. The cells were washed again with the inhibitor-containing PBS. The cell pellets in a tube were suspended in 400 μl hypotonic lysis buffer containing 10 mM KCl, 10 mM HEPES (pH 7.4), 0.05% IGEPAL CA630 (Sigma), 0.2 mM sodium orthovanadate (Sigma), and 1x EDTA-free protease inhibitor cocktail. The suspensions were kept on ice for 20 min and then centrifuged at 14000 rpm for 10 min. The cell pellets with nuclei were suspended in 400 μl extraction buffer consisting of 50 mM Tris-HCl (pH 7.4)/1% Triton X-100/10% Glycerol/150 mM NaCl/25 nM $ZnSO_4$/1 mM $MgCl_2$/0.2 mM sodium orthovanadate/1x EDTA-free protease inhibitor cocktail/50 units/ml benzonase (EMD Millipore), rotated for 30 min, and then centrifuged at 14000 rpm for 10 min. The supernatants (nuclear extracts) typically contained 5 μg/μl protein. This procedure and IP described below were carried out in a cold room.

**Immunoprecipitation.** Nuclear extracts (500 μl) were mixed with 20 μl protein A magnetic beads (NEB), rotated for 1 h, and the supernatants were collected by applying a magnetic field. The precleared nuclear extracts were mixed with 5 μg rabbit ELMSAN1 antibody (Supplementary Table 1) and rotated for 1 h. It was then mixed with 20 μl protein A magnetic beads and rotated further for 75 min. The beads with proteins were collected by applying a magnetic field and washed twice with 500 μl extraction buffer without benzonase and finally washed once with the buffer without both benzonase and Triton X-100. Proteins were eluted from the beads with 30 μl 0.5 M ammonium hydroxide four times. All eluates derived from four tubes ($30 μl \times 4 \times 4$) were combined and lyophilized for MS. The control experiment with water or n-rIgG in place of the antibody was carried out in parallel. For western blotting, the magnetic beads were washed three times with 500 μl extraction buffer without benzonase, suspended in 1.5x loading buffer, and heated for 5 min at 100 °C to elute proteins. Enrichment/purification of protein was calculated by the following equation: (spectral count of a protein/total spectral count of immunoprecipitate)/(spectral count of the protein/total spectral count of nuclear extract). When a protein was detected in the immunoprecipitates but not in the nuclear extracts by MS, spectral count 1 was assigned to the missing protein in the nuclear extracts. Results were also analyzed by SAINT[18].

**SDS-PAGE and western blotting/immunoblotting (IB).** Protein samples were separated on 4–20% Mini-PROTEAN TGX gels (Bio-Rad). Proteins on the gel were stained by the ProtoBlue Safe Colloidal Coomassie G-250 procedure or transferred to a nitrocellulose or a PVDF membrane and then probed by IB. Both gels and blots were scanned and analyzed using LI-COR Odyssey and Image Studio Lite. Protein-bound nitrocellulose/PVDF membranes were incubated with primary antibodies in blocking buffer for 18 h at 4 °C, washed, and incubated with secondary antibodies for 1 h at a room temperature, and thoroughly washed before scanning. For examination of rabbit ELMSAN1 antibody specificity, excess epELM-5 (20 μg) immobilized to a small piece of PVDF membrane was added to the standard IB reaction mixture, and then the regular IB procedure was performed. We prepared the epELM-5 epitope in advance by separating the epitope from bacterial proteins on SDS-PAGE gel, transferring it to PVDF, and finally excising a piece of the PVDF, and prepared the control PVDF using bacterial extracts that the empty vector was expressed.

**Knockdown of genes.** Reverse transfection was performed for knockdown with siRNAs (IDT) and universal negative control 1 according to the protocol of Lipofectamine RNAiMAX (Thermo Fisher), and the cultures were incubated for 3 days. Different siRNAs prepared for a single gene, listed in Supplementary Table 2, knocked down expression of the gene to a similar extent. hs.Ri.MKI67.13.1 and hs.Ri.GNL2.13.2 were used to knockdown genes *MKI67* and *GNL2*, respectively, throughout this study. Small scale cultures were treated with siRNA for knockdown even in experiments in which whole cell lysates and nuclear extracts were subsequently required. For western blotting of whole cell lysates, the culture in a 35-mm dish was briefly rinsed with PBS and harvested with 200 μl of boiling 50 mM Tris-HCl (pH 7.4) with 0.5% SDS. The lysate was mixed with 20 μl 10x

protease inhibitor complete cocktail (with EDTA, Roche) and digested with 1 µl DNase and 1 µl RNase (Roche) in the presence of 10 mM $MgCl_2$ overnight on ice. The sample was mixed with an equal volume of cold 10% trichloroacetic acid (TCA) in acetone and incubated overnight at $-20\,°C$. The precipitated proteins were collected by centrifugation at 14,000 rpm for 10 min, washed with cold acetone, and then dried. The sample was dissolved in 40 µl sample buffer, adjusted to a neutral pH, and heated for 5 min at 100 °C. An aliquot of the resulting sample was applied to a well of 4–20% Mini-PROTEAN TGX gel.

**Protein preparation of MS analysis**. For identification of proteins in nuclear extracts, nuclear extracts were precipitated with a final volume of 12.5% TCA to desalt and remove other compounds, such as Triton X-100, that are found in the extraction buffer. All samples precipitated nuclear extracts, as well as proteins derived from the IPs, were resuspended in 100 µl of 100 mM HEPES, pH 8.5 and digested at 37 °C with trypsin at a 100:1 protein-to-protease ratio overnight. The sample was desalted via StageTip, dried via vacuum centrifugation, and reconstituted in 5% acetonitrile, 5% formic acid for liquid chromatography (LC)–MS/MS processing.

**LC and tandem MS**. Our MS data were collected similar to as described previously[38]. In brief, we used a Q Exactive mass spectrometer (Thermo Fisher Scientific, San Jose, CA) coupled with a Famos Autosampler (LC Packings) and an Accela600 LC pump (Thermo Fisher Scientific). Peptides were separated on a 100 µm inner diameter microcapillary column packed with ~25 cm of Accucore C18 resin (2.6 µm, 150 Å, Thermo Fisher Scientific). For each analysis, we loaded ~1 µg onto the column.

Peptides were separated using a 1 h gradient of 5–25% acetonitrile in 0.125% formic acid with a flow rate of ~300 nl/min. The scan sequence began with an Orbitrap MS1 spectrum with the following parameters: resolution 70,000, scan range 300–1500 Th, automatic gain control (AGC) target $1 \times 10^5$, maximum injection time 250 ms, and centroid spectrum data type. We selected the top 20 precursors for MS2 analysis which consisted of HCD high-energy collision dissociation with the following parameters: resolution 17,500, AGC $1 \times 10^5$, maximum injection time 60 ms, isolation window 2 Th, normalized collision energy 30, and centroid spectrum data type. The underfill ratio was set at 1%, which corresponds to a $1.1 \times 10^4$ intensity threshold. In addition, unassigned and singly charged species were excluded from MS2 analysis and dynamic exclusion was set to automatic.

**Data analysis of tandem MS**. Mass spectra were processed using a Comet-based in-house software pipeline[39]. Spectra were converted to mzXML using a modified version of ReAdW.exe. Database searching was performed using a 50-ppm precursor ion tolerance for total protein-level analysis. The product ion tolerance was set to 0.02 Da. These wide mass tolerance windows were chosen to maximize sensitivity in conjunction with Comet searches and linear discriminant analysis[39,40]. Oxidation of methionine residues (+15.995 Da) was set as a variable modification. Peptide-spectrum matches (PSMs) were adjusted to a 1% FDR[41,42]. PSM filtering was performed using a linear discriminant analysis, as described previously[39], while considering the following parameters: XCorr, ΔCn, missed cleavages, peptide length, charge state, and precursor mass accuracy. PSMs were identified, quantified, and collapsed to a 1% peptide FDR and then collapsed further to a final protein-level FDR of 1%. Moreover, protein assembly was guided by principles of parsimony to produce the smallest set of proteins necessary to account for all observed peptides.

**Immunofluorescence staining of cell cultures**. Growing HEK293T cultured in a Matsunami Glass Bottom Dish (D11130H) was rinsed with room temperature 1x PBS and then incubated in PBS/4% paraformaldehyde, pH 7.2, for 10 min at room temperature. The fixation was stopped with 3x PBS, followed by replacement with 1x PBS, then successively dehydrated with 50, 75, 95, and 100% ethanol, and eventually air-dried. These fixed cultures were permeabilized in 1x PBS/0.2% Triton X-100 for 10 min at room temperature and blocked with 1x PBS/5% BSA/ 0.1% IGEPAL CA630 for 5 min. These ready-to-go samples were incubated with two primary antibodies (Supplementary Table 1) in PBS/BSA/IGEPAL CA630 overnight (16–18 h) at 4 °C. They were washed four times with PBS/BSA/IGEPAL CA630, incubated with two secondary goat antibodies (anti-rabbit and anti-mouse IgGs) in PBS/BSA/IGEPAL CA630 for 1 h at room temperature, washed three times and rinsed with water, and air-dried. Subsequently, these samples were treated with DAPI-containing Vectashield (Vector) and then inspected using a Yokogawa CSU-X1 spinning disk confocal on an inverted Nikon Ti fluorescence microscope in the Nikon Imaging Center at Harvard Medical School. Immunofluorescent image was taken by capturing emission derived from 300 nm excitation of both Alexa Fluor 555 (red) and Alexa Fluor 488 (green) dyes and emission derived from 100 ms excitation of DAPI. Z-plane images of a cell (or a culture field) were taken every 0.25 µm, and these images slicing the nucleolus through the center were used for comparison. DAPI has more than 50-fold and more than 100-fold stronger affinity for polydA-polydT than poly [(d(G-C))2] and polyA-polyU, respectively, and stronger emission with polydA-polydT than the other two[32].

**Statistics and reproducibility**. Independent culture result was presented as a dot, and data between two groups were compared by two-tailed Student's *t* test. *P* value < 0.05 indicates significant difference between the two groups. GraphPad Prism was used for the *t*-test, and it was also used for correlation efficient analysis of proteins between HEK293T and Hela.cl1.

**Analytical software and database**. Websites that we used for data analysis:
GenBank database at National Center for Biotechnology Information was used for various computational analyses of proteins (https://blast.ncbi.nlm.nih.gov/Blast.cgi?PROGRAM=blastp&PAGE_TYPE=BlastSearch&LINK_LOC=blasthome) and nucleic acids (https://www.ncbi.nlm.nih.gov/nuccore/).
ImageJ (https://imagej.nih.gov/ij/download.html) was used for analysis of immune-stained images. The human UniProt database (https://www.uniprot.org) was used for identification of proteins analyzed by MS.
SAINT (https://reprint-apms.org/?q=analysis_front_apms).
Cytoscape (https://cytoscape.org/download.html) and the STRING database were used for analysis of protein–protein interaction.
PrDOS (https://prdos.hgc.jp/cgi-bin/top.cgi) was used for IDPs prediction.
Lasergene (DNASTAR) was used for sequence analysis.

**Reporting summary**. Further information on research design is available in the Nature Research Reporting Summary linked to this article.

## Data availability

The MS proteomics data have been deposited to the ProteomeXchange Consortium via the PRIDE[43] partner repository with the dataset identifier PXD024234. Supplementary Data file includes five files: file 1, inhibition of nuclear protein binding to the *IGFBPL1* promoter sequence by the CPN domain cluster; file 2, enrichment of proteins by IP with the anti-ELMSAN1 antibody; file 3, SAINT analysis of the IP–MS data (#1 and #2); file 4, SAINT analysis of another duplicate IP–MS data (#3 and #4); and file 5, comparison of RNAmetasome constituent proteins between HEK293T and Hela.cl1. Supplementary Information file includes Supplementary Figs. 1–13 (uncropped IB images) and Supplementary Tables 1 and 2 for "Methods" and Table 3 for quantitation of reciprocal IP yield. The plasmid constructed in this work, pET424 epELM-5, is available from Addgene, ID: 179391. All other data are available from the corresponding author on reasonable request.

## Code availability

Mass spectra were acquired using a QQ Exactive HF-X mass spectrometer instrument from Thermo Fisher Scientific with the corresponding software (Tune 2.9 QF1) provided by the vendor. Raw files were converted to mzXML using Raw File Reader (v3.0.77) provided by Thermo Fisher Scientific. Spectra were searched using SEQUEST 28. Search results were filtered using the LDA function in MASS Package in R as described[39]. All figures were made using Excel 2013 or R 3.4.2.

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

## Acknowledgements

The authors would like to thank Dr. Danesh Moazed for all his support, Dr. Steven Gygi for providing access to MS facilities, and the Nikon Imaging Center at Harvard Medical School for help with light microscopy. This work has been achieved by a private fund from late Dr. Howard Green to S.I. J.A.P was supported by NIH Grant GM132129.

## Author contributions

S.I. designed the study, performed experiment except for MS, analyzed data, and wrote the manuscript. J.A.P. performed MS and SAINT analysis, wrote MS methods, and edited the manuscript.

## Competing interests

The authors declare no competing interests.
