## [Transparent Peer Review File · Communications Biology]

Referee expertise:

Referee #1:

Referee #2:

Referee #3:

Reviewers' comments:

Reviewer #1 (Remarks to the Author):

In this manuscript Iuchi and Paulo identify a new complex, composed of numerous RNA processing proteins in Hek293T cells that they term the RNAmetasome. This complex was identified by mass-spec using ELMSAN1 as bait. Reciprocal IPs were used to validate a number of the mass spec hits. I do not question the results of the mass spec analysis, however the authors make many grand claims in the paper that are not substantiated with any sort of biological data. While the mass spec data in this paper may support a hypothesis that the RNAmetasome is involved in several aspects of RNA metabolism, this hypothesis needs experimental validation, for which there is none in this paper.

Concerns:

1. The introduction is confusing and does not sufficiently set up the research question. This manuscript needs a much better description of the key players/proteins discussed in the paper and why they are important. For example, it's unclear to this reviewer what is the significance of ELMSAN1. A pubmed search returns only a single hit for this (Mondal et al eLife 2020).
2. There is no experimental validation of the RNAmetasome and its proposed biological role.
3. The authors compare differences to other published IP's and suggest that RNA may be an important factor for protein interactions. For a complex proposed to function in RNA metabolism it seems of utmost importance that all of the mass spec and reciprocal IPs be repeated without RNase in the buffer so that conclusions can be more appropriately drawn about the role of RNA in mediating interactions within the RNAmetasome.
4. Throughout the manuscript abbreviations and introductions for proteins are needed.
5. The siRNAs knockdowns in Fi. 2C are very poor and too many conclusions are drawn from this. Why are there multiple bands for ELMS? The sequences for the siRNAs must be provided in the methods.
6. Lack of an interaction by co-IP does not necessarily mean that things do not interact. For example, the authors claim that ELMSAN1 does not bind HDAC1, HDAC2, H3, and H2B. This could be a weak or transient interaction or something critical for binding may be missing in the IP buffer. On this same line the lack of detecting a protein by mass spec does not mean that it's not there.
7. What is the axis in Fig. 1C? MW ladder is missing in Fig 1D.

Reviewer #2 (Remarks to the Author):

Shiro Iuchi and Joao A. Paulo describe characterization of an "RNAmetasome" network of ELMSAN1-interacting proteins and propose a model for regulation of RNAmetasome binding to chromatin. Although the work presents interesting insights into the composition of ELMSAN1 network, it requires major revisions prior to publication in Communications Biology to improve clarity and to provide additional support to the main conclusions. The suggestions are outlined below:

1. Extended Data Fig. 1: It is unclear how "+CPN and -CPN" experiment was set up, please add the description of the experimental setup in the Results and/or in the Methods sections.
2. Page 3, line 54: Please clarify how "protein concentration between after and before binding" was

determined in the immunoaffinity purification (IP) experiments.

3. Page1, line 14, page2, line 42 and in other parts of the manuscript: "huge protein complex" description is vague. To provide scientific support for this claim, word "huge" can be replaced by the actual estimated size of the complex or the number of its components.

4. Please define "ELMSAN1" in the introduction

5. Fig. 1c: please label the y-axis

6. Fig. 1 and Extended Data Fig.3 IP experiments: Results from these experiment have to be consolidated, normalized and processed into a single dataset to be plotted on the same figure. A well-established proteomic software can be used for consolidation of proteomics data (e.g., SAINT(Choi et al., nat. Methods, 2011), MSstats(Choi et al., Bioinformatics, 2014)). As is, it is unclear how well the data is replicated in these two analyses, which complicates data interpretation.

7. P6, line 138: Statement "... a huge protein complex is involved with diverse RNA and its downstream metabolism. Therefore, we named this complex "RNAmetasome"." does not have enough evidentiary support from this study. The authors characterized an interaction network of ELMSAN1 protein that includes several highly connected clusters assigned based on the STRING interactions database. However, there is no evidence that these interactions occur in the same place and time to define them as a "complex". These interactions can be transient and occur in different parts of the nucleus with a different subset of identified proteins. In addition, some of them can be artifacts of cell lysis prior to immunoaffinity purification. The authors should revise the name "RNAmetasome complex" to "RNAmetasome network" throughout the manuscript or provide additional evidence that a multisubunit complex does exist, such as proximity labeling experiments, immunofluorescence studies, reciprocal IP studies, isolation and structural analysis of this protein complex, etc. If the authors consider the complex of five proteins defined in Fig.3h as an "RNAmetasome complex", this should be clarified throughout the manuscript.

8. Extended Data Fig5: the image size is too small. Please replace with a high-quality image.

9. Figure 4, panels 42, 43, 46, and 47: please check that the labels correspond with the staining, in particular is MKI67 staining red or green?

10. Page15, line 293: please correct "DNAP" to "DAPI"

11. Extended Data Fig.7: this model is based on the immunofluorescence staining of the proteins in the presence and absence of GNL2 (Fig.4 and Extended Data Fig.6). Additional evidence, such as quantitation of the observed changes in protein localization upon GNL2 KD and/or co-IP experiments from wild type cells and GNL2 KD cells will provide a stronger support for the model.

12. Please specify the database source and version used for searching MS data in Sequest in "Methods."

Reviewer #3 (Remarks to the Author):

The manuscript by Iuchi and Paulo describes an extensive network of interactions between proteins involved in various stages of gene expression in 293T cell line. The authors claim that hundreds of proteins form a huge multiprotein complex, which they coin "RNAmetasome". Overall, there are some potentially interesting novel aspects of the findings made by the authors, however at the present form the manuscript contains many overstatements and the data is somewhat superficial.

While studying the mechanism underlying inhibition of ELMSAN1 binding to CpG-rich elements by KDM2A demethylase, the authors noted that ELMSAN1 interacts with a different set of proteins than reported previously by others.

Major remarks:

1. Iuchi and Paulo performed their studies in just one cell line, namely HEK293T. It is different from cell lines used by Itoh et al. (2015) and particularly Bantscheff et al. (2011), which precludes direct comparison of the data. To see how universal is the network of interactions described by the authors, they are encouraged to validate it in several other cell lines of various origins.

2. The authors state that HDACs and histones are absent in ELMSAN1 co-immunoprecipitates. However, in the Extended Data Fig. 2b, faint bands corresponding to HDAC1, H2B and H3 are present.

I agree that anti-ELMSAN antibodies recover much more DNTTIP1 than HDACs or histones, but I would suggest to mitigate a bit the authors' claim that ELMSAN1 does not interact with the latter proteins in 293T cells. Rather, the stoichiometry of the interaction might be different from the one reported previously.

Could the authors present longer exposures of the WB data shown in the Extended Data Fig. 2b?

3. I am afraid I do not understand the authors' conclusion from the RNAi experiment shown in the Extended Data Fig. 2c. "Knockdown of the ELMSAN1 gene with siRNAs diminished the 140-kDa protein, suggesting that our result is the consequence of ELMSAN1 IP". I agree that the WB shows that each of the 3 siRNAs utilized led to modest (~50% on average) reduction of the band corresponding to ELMSAN1. However, this WB also uncovers the major technical flaw of the manuscript, which is related to extremely low specificity of anti-ELMSAN1 antibodies (based on the results presented in the Extended Data Fig. 2c it is evident that these antibodies cross react with at least 16 other proteins). In the subsequent sections of the paper, the authors focus on proteins interacting with ELMSAN1, i.e. identified as enriched in anti-ELMSAN1 co-IP by mass-spec analysis. Taking into account how unspecific the antibodies are, how could they rule out that what they observe is not due to the recognition of completely unrelated proteins by the anti-ELMSAN1 antibodies?

4. Fig. 1d – the amounts of many proteins shown in this figure are indeed higher for co-IP with anti-ELMSAN1 antibodies than for control IgG. However, the quantitative differences are not particularly impressive. Could the authors perform WB analysis for their second data set, try to calculate the enrichment folds and match them with quantitative analyses from MS experiments?

5. Throughout the manuscript (including the title and the abstract) I would suggest to tone down the statement about the existence of a novel multiprotein complex, consisting of almost 500 (!) components. I would say that based on authors' results some novel elements of a large interaction network between proteins participating in different phases of gene expression regulation begin to emerge.

6. Functional analyses that would support the significance of the authors' findings are missing. I would suggest to focus on the "RNAmetasome core complex" defined by the authors and see for instance what would be the consequence of MKI67 scaffolding protein depletion on different aspects of RNA metabolic processes, to which this "core" appears to be connected, such as e.g. rRNA processing, mRNA degradation, decapping, quality control. The authors say about the involvement of RNAmetasome network into the biogenesis of macromolecules, but they neither specify nor demonstrate what kind of macromolecules they have in mind.

7. I appreciate the reciprocal co-IP experiments validating protein interactions within ELMSAN1/MDN1/MKI67/YTHDC2/GNL2 subcomplex. It is now absolutely critical that the authors use these coimmunoprecipitates for mass-spec analysis and compare the results to those obtained with anti-ELMSAN1 antibody. Perhaps this would allow to limit the number of significant "RNAmetasome" network components to some reasonable value and make the data more meaningful and reliable.

8. The immunostaining approach applied by the authors nicely showed the co-localization of the proteins of interest, but still it does not prove that they interact within the cell. More sophisticated analyses like BiFC or FLIM-FRET are required. Likewise, experiments with siRNA-mediated protein depletion, resulting in dysregulation of other RNAmetasome components, modified nucleolar morphology and subunit re-localization do not provide sufficient evidence at this stage to justify the statement that the identified protein-protein interaction network participates in ribosome biogenesis. Additional experiments (e.g. polysome profiling, analysis of pre-rRNA processing paths, analyses of ribosome assembly) are required to decipher the connection between RNAmetasome and ribosome synthesis.

Minor comments:

- Abstract: "RNA plays a central role in macromolecule biogenesis, participating in various pathways, such as expression and processing of genetic information". This sentence is quite strange – please, rephrase it.

- Abstract: "However, RNA must be converted from nascent to functional forms for that role". I see no connection between this clause and the manuscript content.

- Abstract: "It consists of hundreds of proteins involved in RNA and its downstream metabolism."

Some word (synthesis?) is missing after "RNA"?

- line 41: correct "ELMSNA1"

- line 64: correct "ELMANSAN1"

- line 134 and elsewhere: the term "EXOSC complex" is not widely recognized in the literature. I would suggest to change it to "RNA exosome"

- line 463: correct "elutes"

ONE-BY-ONE RESPONSE TO REVIEWERS' COMMENTS, WHICH IS WRITTEN IN BLUE.

Reviewer #1 (Remarks to the Author):

In this manuscript Iuchi and Paulo identify a new complex, composed of numerous RNA processing proteins in Hek293T cells that they term the RNAmetasome. This complex was identified by mass-spec using ELMSAN1 as bait. Reciprocal IPs were used to validate a number of the mass spec hits. I do not question the results of the mass spec analysis, however the authors make many grand claims in the paper that are not substantiated with any sort of biological data. While the mass spec data in this paper may support a hypothesis that the RNAmetasome is involved in several aspects of RNA metabolism, this hypothesis needs experimental validation, for which there is none in this paper.

Concerns:

1. The introduction is confusing and does not sufficiently set up the research question. This manuscript needs a much better description of the key players/proteins discussed in the paper and why they are important. For example, it's unclear to this reviewer what is the significance of ELMSAN1. A pubmed search returns only a single hit for this (Mondal et al eLife 2020).

Thank you for bringing up the publication by Mondel et al. It certainly has important information and was helpful to us. We have improved the introduction adding information of ELMSAN1 and some other information. We have also added two more phrases to the introduction to clarify the logic.

2. There is no experimental validation of the RNAmetasome and its proposed biological role.

We performed an additional IP-MS and IP-IB with HEK293T. The result revealed more details of the RNAmetasome network. Results with Hela.c11 cells also showed that the RNAmetasome network is present. These 2 additional experiments validate the presence of RNAmetasome network in human cells. As the RNAmetasome network is supposed to eventually link to cell proliferation, we did an experiment by gene knockdown to show how it affects cell proliferation. The answer is more complicated than just "yes". This is a similar situation to gene silencers that often work the other way around as well. We discussed the results including the effect as observed by cytology.

3. The authors compare differences to other published IP's and suggest that RNA may be an important factor for protein interactions. For a complex proposed to function in RNA metabolism it seems of utmost importance that all of the mass spec and reciprocal IPs be repeated without RNase in the buffer so that conclusions can be more appropriately drawn about the role of RNA in mediating interactions within the RNAmetasome.

Our new duplicate IP-MS experiment identified EZH2, SUS12, and JARID2 of PRC1 and 2 complexes at a subtle level. This result is consistent with result by Zhou et al who did their IP without digestion of RNA. As we wish to observe protein-protein interaction networks, RNA must be removed from our assay buffer.

4. Throughout the manuscript abbreviations and introductions for proteins are needed.

We have improved the text as much as possible and added the website address (<https://www.uniprot.org/>) for readers to refer to it.

5. The siRNAs knockdowns in Fi. 2C are very poor and too many conclusions are drawn from this. Why are there multiple bands for ELMS? The sequences for the siRNAs must be provided in the methods.

We have added new results (Figure 1e,f) concerning the specificity of the antibody. The antibody is specific. Five ELMSAN1 isomers are now known, and they may be partially degraded. As a result, those would make up many bands. That is why so many bands were identified.

Sequence of the siRNAs have been provided in Supplementary Table 3.

6. Lack of an interaction by co-IP does not necessarily mean that things do not interact. For example, the authors claim that ELMSAN1 does not bind HDAC1, HDAC2, H3, and H2B. This could be a weak or transient interaction or something critical for binding may be missing in the IP buffer. On this same line the lack of detecting a protein by mass spec does not mean that it's not there.

You are absolutely right. We have revised the original statements.

7. What is the axis in Fig. 1C? MW ladder is missing in Fig 1D.

The axis shows enrichment, which has been added. The MW has been added. Please note that the figure (Fig1d) has been renamed Supplementary Figure 2d.

Reviewer #2 (Remarks to the Author):

Shiro Iuchi and Joao A. Paulo describe characterization of an “RNAmetasome” network of ELMSAN1-interacting proteins and propose a model for regulation of RNAmetasome binding to chromatin. Although the work presents interesting insights into the composition of ELMSAN1 network, it requires major revisions prior to publication in Communications Biology to improve clarity and to provide additional support to the main conclusions. The suggestions are outlined below:

1. Extended Data Fig. 1: It is unclear how “+CPN and –CPN” experiment was set up, please add the description of the experimental setup in the Results and/or in the Methods sections.

We have described the experimental procedure in the text. We have renamed Extended Data Fig. 1 to Figure 1a and added the description to the text and the legend. We also have added a table of the result (Supplementary Table 1).

2. Page 3, line 54: Please clarify how “protein concentration between after and before binding” was determined in the immunoaffinity purification (IP) experiments.

The description has been clarified by equations.

3. Page1, line 14, page2, line 42 and in other parts of the manuscript: “huge protein complex” description is vague. To provide scientific support for this claim, word “huge” can be replaced by the actual estimated size of the complex or the number of its components.

All instances of “huge” were replaced with “hundreds”.

4. Please define “ELMSAN1” in the introduction.

We have introduced ELMSAN1 in the new version.

5. Fig. 1c: please label the y-axis

The axis is labelled with Enrichment.

6. Fig. 1 and Extended Data Fig.3 IP experiments: Results from these experiment have to be consolidated, normalized and processed into a single dataset to be plotted on the same figure. A well-established proteomic software can be used for consolidation of proteomics data (e.g., SAINT(Choi et al., nat. Methods, 2011), MSstats(Choi et al., Bioinformatics, 2014)). As is, it is unclear how well the data is replicated in these two analyses, which complicates data interpretation.

According to your advice, we analysed the data by SAINT while we keep the original analysis. The SAINT output basically gave the same conclusion, accompanying more details, and provided statistical relevance. Further, we did another round duplicate IP-MS experiment and also analysed the data by SAINT. This new result agreed with the original conclusion and added more functional groups. These all data have been included in the new version, as Figure 2, Supplementary Figures 4 and 5, and Supplementary Tables 4-6.

7. P6, line 138: Statement "... a huge protein complex is involved with diverse RNA and its downstream metabolism. Therefore, we named this complex "RNAmetasome"." does not have enough evidentiary support from this study. The authors characterized an interaction network of ELMSAN1 protein that includes several highly connected clusters assigned based on the STRING interactions database. However, there is no evidence that these interactions occur in the same place and time to define them as a "complex". These interactions can be transient and occur in different parts of the nucleus with a different subset of identified proteins. In addition, some of them can be artifacts of cell lysis prior to immunoaffinity purification. The authors should revise the name "RNAmetasome complex" to "RNAmetasome network" throughout the manuscript or provide additional evidence that a multisubunit complex does exist, such as proximity labeling experiments, immunofluorescence studies, reciprocal IP studies, isolation and structural analysis of this protein complex, etc. If the authors consider the complex of five proteins defined in Fig.3h as an "RNAmetasome complex", this should be clarified throughout the manuscript.

I completely agree with your argument. We have replaced "RNAmetasome" with "RNAmetasome network" in the new text.

8. Extended Data Fig5: the image size is too small. Please replace with a high-quality image.

It has been enlarged. Further, a series of images for ELMSAN1 were replaced with clearer ones. This Fig. has been renamed Supplementary Figure 6.

9. Figure 4, panels 42, 43, 46, and 47: please check that the labels correspond with the staining, in particular is MKI67 staining red or green?

Yes, the panels were in wrong positions. They have been corrected.

10. Page15, line 293: please correct "DNAP" to "DAPI"

DNAP has been corrected to DAPI

11. Extended Data Fig.7: this model is based on the immunofluorescence staining of the proteins in the presence and absence of GNL2 (Fig.4 and Extended Data Fig.6). Additional evidence, such as quantitation of the observed changes in protein localization upon GNL2 KD and/or co-IP experiments from wild type cells and GNL2 KD cells will provide a stronger support for the model.

We did co-IP-IB with the anti-MDN1 antibody for GNL2 KD cultures. MDN1 can be free of GNL2 but MKI67 was still co-immunoprecipitated. We think these 2 proteins interact with each other unless the two were physically separated. The figure has been improved by gathering information from other results and renamed Figure 5.

12. Please specify the database source and version used for searching MS data in Sequest in “Methods.”

We have specified the database source (the ProteomeXchange Consortium, ID: PXD024234) and the information on the version used for searching MS data (Comet)

Reviewer #3 (Remarks to the Author):

The manuscript by Iuchi and Paulo describes an extensive network of interactions between proteins involved in various stages of gene expression in 293T cell line. The authors claim that hundreds of proteins form a huge multiprotein complex, which they coin “RNAmetasome”. Overall, there are some potentially interesting novel aspects of the findings made by the authors, however at the present form the manuscript contains many overstatements and the data is somewhat superficial.

While studying the mechanism underlying inhibition of ELMSAN1 binding to CpG-rich elements by KDM2A demethylase, the authors noted that ELMSAN1 interacts with a different set of proteins than reported previously by others.

Major remarks:

1. Iuchi and Paulo performed their studies in just one cell line, namely HEK293T. It is different from cell lines used by Itoh et al. (2015) and particularly Bantscheff et al. (2011), which precludes direct comparison of the data. To see how universal is the network of interactions described by the authors, they are encouraged to validate it in several other cell lines of various origins.

We have added analysis of another cell line, Hela.c11. The result shows that the RNAmetasome network exists in the cell line as well (Figure 6a). By extending experiments with this cell line, we have provided further evidence that MDN1 and MKI67 controls the formation of the RNAmetasome network (Figure 6).

2. The authors state that HDACs and histones are absent in ELMSAN1 co-immunoprecipitates. However, in the Extended Data Fig. 2b, faint bands corresponding to HDAC1, H2B and H3 are present. I agree that anti-ELMSAN1 antibodies recover much more DNMT1 than HDACs or histones, but I would suggest to mitigate a bit the authors' claim that ELMSAN1 does not interact with the latter proteins in 293T cells. Rather, the stoichiometry of the interaction might be different from the one reported previously.

Could the authors present longer exposures of the WB data shown in the Extended Data Fig. 2b?

We did longer exposure, but it did not improve the result. We also did the IP-IB extending the incubation time of the IP from 2h 15min to overnight, and somewhat observed co-IP of HDAC2 but not that of HDAC1 (we have not shown the result in this work). Two duplicate IP-MS experiments found HDAC2 coimmunoprecipitated better than HDAC1 (Supplementary Table 5-6). In any case, these co-IP were repeatable but at a low level.

I agree with your argument and have changed the expression in the new version from "ELMSAN1 bind neither HDACs nor histones" to "ELMSAN1 does not appreciably bind HDACs and histones".

3. I am afraid I do not understand the authors' conclusion from the RNAi experiment shown in the Extended Data Fig. 2c. "Knockdown of the ELMSAN1 gene with siRNAs diminished the 140-kDa protein, suggesting that our result is the consequence of ELMSAN1 IP". I agree that the WB shows that each of the 3 siRNAs utilized led to modest (~50% on average) reduction of the band corresponding to ELMSAN1. However, this WB also uncovers the major technical flaw of the manuscript, which is related to extremely low specificity of anti-ELMSAN1 antibodies (based on the results presented in the Extended Data Fig. 2c it is evident that these antibodies cross react with at least 16 other proteins). In the subsequent sections of the paper, the authors focus on proteins interacting with ELMSAN1, i.e. identified as enriched in anti-ELMSAN1 co-IP by mass-spec analysis. Taking into account how unspecific the antibodies are, how could they rule out that what they observe is not due to the recognition of completely unrelated proteins by the anti-ELMSAN1 antibodies?

Although this antibody may recognize something else besides canonical ELMSAN1 by WB, 5 isoforms of the protein are reported to the UniProt database, and it is likely that partially degraded isoforms are also included in the extracts. Accordingly, many proteins should be detected with a good antibody. In addition, as the LI-COR Odyssey is extremely good at long range protein quantitation, our image of WB faithfully

shows what and how many proteins are present. We did a few experiments to evaluate the anti-ELMSAN1 antibody and found that the antibody is specific and suitable for WB (Figure 1d-f). Therefore, our IP-MS data are reliable.

4. Fig. 1d – the amounts of many proteins shown in this figure are indeed higher for co-IP with anti-ELMSAN1 antibodies than for control IgG. However, the quantitative differences are not particularly impressive. Could the authors perform WB analysis for their second data set, try to calculate the enrichment folds and match them with quantitative analyses from MS experiments?

We did another round IP-MS including two kinds of controls with water and normal IgG and analysed the result by SAINT (Supplementary Table 6). The outcome is better than the original conclusion (Supplementary Figure 5).

This high background with the normal IgG occurs with the HEK293T extracts but not with the Hela.c11 extracts (Supplementary Figure 8). We don't know why it happens only with the HEK293T extracts. However, the high background is likely to show involvement of protein phase separation, which directs proteins to interact with other proteins without specific domains for the formation of the RNAmetasome network. This possibility has been discussed in the second paragraph of the discussion section.

5. Throughout the manuscript (including the title and the abstract) I would suggest to tone down the statement about the existence of a novel multiprotein complex, consisting of almost 500 (!) components. I would say that based on authors' results some novel elements of a large interaction network between proteins participating in different phases of gene expression regulation begin to emerge.

I agree with you, and we replaced the expression "RNAmetasome" with "RNAmetasome network"

6. Functional analyses that would support the significance of the authors' findings are missing. I would suggest to focus on the "RNAmetasome core complex" defined by the authors and see for instance what would be the consequence of MKI67 scaffolding protein depletion on different aspects of RNA metabolic processes, to which this "core" appears to be connected, such as e.g. rRNA processing, mRNA degradation, decapping, quality control. The authors say about the involvement of RNAmetasome network into the biogenesis of macromolecules, but they neither specify nor demonstrate what kind of macromolecules they have in mind.

The RNAmetasome network suggests that the macromolecules that the network processes should be mature RNAs, proteins, and ribosomes; and ultimate outcome of the processing would be cell proliferation. Accordingly, we did knockdowns of

MKI67 and GNL2. The result is not simple: MKI67 KD did not slow down the cell proliferation, but GNL KD did (Figure 7). MKI67 does not appear to have a simple stimulatory role for growth. This is a similar situation to cases of regulators that activate a group of genes and repress another group of genes. We discussed role of the core proteins in the discussion, especially in paragraphs 1 (lower half) and the last paragraph.

7. I appreciate the reciprocal co-IP experiments validating protein interactions within ELMSAN1/MDN1/MKI67/YTHDC2/GNL2 subcomplex. It is now absolutely critical that the authors use these coimmunoprecipitates for mass-spec analysis and compare the results to those obtained with anti-ELMSAN1 antibody. Perhaps this would allow to limit the number of significant “RNAmetasome” network components to some reasonable value and make the data more meaningful and reliable.

I agree with you. That is good at determining common factors. However, that is not good at determining a limit of the system with the anti-ELMSAN1 antibody. The duplicate IP-MS results that we added have increased reliability of our data. The reliability is offered by enrichment of distal proteins, such as SUZ12, EED, and JARID2, that were recently shown to interact with rixosome proteins by another group’s IP.

8. The immunostaining approach applied by the authors nicely showed the co-localization of the proteins of interest, but still it does not prove that they interact within the cell. More sophisticated analyses like BiFC or FLIM-FRET are required. Likewise, experiments with siRNA-mediated protein depletion, resulting in dysregulation of other RNAmetasome components, modified nucleolar morphology and subunit re-localization do not provide sufficient evidence at this stage to justify the statement that the identified protein-protein interaction network participates in ribosome biogenesis. Additional experiments (e.g. polysome profiling, analysis of pre-rRNA processing paths, analyses of ribosome assembly) are required to decipher the connection between RNAmetasome and ribosome synthesis.

Thank you for the suggestion. We would attempt it at the next steps, and have suggested this as future experiments in the discussion.

Minor comments:

- Abstract: “RNA plays a central role in macromolecule biogenesis, participating in various pathways, such as expression and processing of genetic information”. This sentence is quite strange – please, rephrase it.

We have improved it.

- Abstract: “However, RNA must be converted from nascent to functional forms for that role”. I see no connection between this clause and the manuscript content.

We improved it by changing the order of words of the next sentence, so that the new description should be more friendly to readers.

- Abstract: “It consists of hundreds of proteins involved in RNA and its downstream metabolism.”. Some word (synthesis?) is missing after “RNA”?

The sentence has been replaced with “In HEK293T, RNAmetasome network consists of proteins responsible for gene expression, splicing, ribosome biogenesis, chromatin remodelling, and cell cycle.”

- line 41: correct “ELMSNA1”

Corrected

- line 64: correct “ELMANSAN1”

Corrected

-

line 134 and elsewhere: the term “EXOSC complex” is not widely recognized in the literature. I would suggest to change it to “RNA exosome”

We have corrected it

- line 463: correct “elutes”

Corrected to eluants

Reviewers' comments:

Reviewer #1 (Remarks to the Author):

While the authors have improved the manuscript to some extent, my original assessment that the article is speculative and lacking in biological validation still stands. Co-localization studies and poorly controlled cell growth experiments are not sufficient to convince me of a biological role of the RNA metasome network. Without experimental validation the mass-spec and IP results revealing the RNA metasome network are more well suited for a proteomics specific journal. Moreover, many of the experiments continue to have issues, such as poor siRNA knockdown, non-specific binding, etc.

Reviewer #2 (Remarks to the Author):

Additional experiments were helpful in strengthening authors' conclusions. However, the manuscript requires proofreading and clarifications, a few examples are provided below:

Line 43: In the phrase "spec of the network" what is "spec"?

Line 59: Confusing grammar: "As such, RNA metabolisms are interwound and complicated, but its role can be summarized..."

Line 76: ELMSNAN1 misspelled

Line 88: "The experiment was conducted as follows." This sentence does not connect to the previous description of KDM2A.

Line 683: What is "typical" protein?

Line 111-113: This sentence needs rephrasing to clarify the point: "Likewise, epELM-512 inhibited the antibody from recognising most other bands, even though epELM-5 liberated from PVDF and bound to the protein bands and the backgrounds made the inhibition look weak."

Line 126: What is "converse relationship"? Do the authors mean "inverse"?

Line 307: Capitalize "We"

Reviewer #3 (Remarks to the Author):

In their revised manuscript, Iuchi and Paulo have toned down their claim about the identification of a novel multiprotein "complex" and adequately addressed majority of the concerns that I had after reading the initial version of the paper. Specifically, they verified the existence of the RNA metasome network in another human cell line, namely HeLa, and documented that MDN1 and MKI67 proteins are primarily responsible for the network assembly in different cell lines. Further, they performed additional western-blot analyses focused on interaction between ELMSAN1 and HDAC and histones and modified their claims accordingly to newly obtained results. Specificity of the anti-ELMSAN1 antibodies has been now more reliably proven through additional experiments employing inhibition of protein recognition with ELMSAN1-derived epitope. This makes the coIP-MS data more trustworthy. New set of coIP-MS analysis was also performed to better assess quantitative differences between particular samples. I feel that the results presented in this communication are a good starting point for a comprehensive characterization of the functional importance of the RNA metasome network in a variety of cellular processes. Together with amendments and clarifications in the text, the manuscript by Iuchi and Paulo has been improved and I am now willing to recommend accepting it for publication in Communications Biology.

Minor comments that should be addressed prior to final acceptance:

- extensive editing of English would be desirable, since multiple fragments of the manuscript (particularly in the Results section) are not written clearly enough (e.g. lines 123-127; 180-183); several suggestions for minor text corrections are provided below;

- what is the exact meaning of the following sentence in the abstract: "However, inherent attenuation 43 of MDN1 and MKI67 diminishes its spec of the network"? Please, rephrase it to make it more understandable. Also, merge the information about existence of the RNAmetasome network in different human cell lines (the sentence "Hela.cl1 also has the RNAmetasome network" must be removed);
- The authors must clearly indicate the source of HeLa.cl1 cell line (e.g. is it available from ATCC?)
- line 59: "RNA metabolisms" – change into "RNA metabolic processes"
- lines 60-64 – merge into one sentence, since some information is unnecessarily repeated
- line 70: "regulator" – change into "regulatory"
- line 80: remove comma after "and"
- line 94: "inhibits ELMSAN1 from binding the IGFBPL1 promoter sequence" – change into inhibits ELMSAN1 binding to the IGFBPL1 promoter sequence"
- lines 105-106: "Therefore, the IP result is indeed valid" – remove this sentence completely or move it after arguments proving that IP results are reliable
- line 107 and elsewhere: "DsiRNAs" – change into "siRNAs"
- line 169: "changes" – convert into "change"
- first paragraph of the Discussion should be incorporated into the results section; accordingly, Figure 6 and associated descriptions should be placed earlier in the results section

RESPONSE TO REVIEWERS' REMARKS, WRITTEN IN BLUE

Reviewer #1 (Remarks to the Author):

While the authors have improved the manuscript to some extent, my original assessment that the article is (1) speculative and lacking in biological validation still stands. (2) Co-localization studies and (3) poorly controlled cell growth experiments are not sufficient to convince me of a biological role of the RNAmetasome network. (4) Without experimental validation the mass-spec and IP results revealing the RNA metasome network are more well suited for a proteomics specific journal. Moreover, many of the experiments continue to have issues, such as (2) poor siRNA knockdown, (5) non-specific binding, etc.

Reviewer #1 provides many issues without pointing out specific problems. So, I have tagged these issues with blue-color numbers to respond to each issue.

- (1) Review #1 tries to paint our investigation as a meaningless work by saying that the article is speculative and lacking biological validation as if our manuscript contained only speculations. We respectfully disagree. We present results and conclusions, distinctively separating facts and speculations, both of which are important in a scientific paper. Please see below for the reasons why I disagree with these remarks on the specific issues that this reviewer has raised.
- (2) Our knockdown results of siRNA knockdown verify the involvement of a specific gene and its specific isomer protein. siRNAs cause a wide and strong off-target effect at higher than 50 nM concentration and can cause it even at 25 nM (Persengiev. RNA (2004); Sigoillot. Nat Methods (2012); Neumeier; Frontiers in Plant Science (2021). To avoid such complication, we used siRNAs at 10 to 40 nM concentration throughout our experiments. In the case of the *ELMSAN1* knockdown, we used three different siRNAs at 10 nM, and three of them diminished 140-kDa ELMSAN1 to a similar extent. This is how we determined the 140-kDa protein is canonical ELMSAN1. Moreover, protein bands including the 140-kDa protein band were characterized as proteins relevant to ELMSAN1 by inhibition with an engineered ELMSAN1 epitope. The evidence clears the reviewer's doubt. In the cases of *MKI67* and *GNL2*, we used two different siRNA in the beginning of the experiments and observed that the two had a similar effectiveness, and then used one of the two siRNA at 10-40 nM throughout the experiments. These experimental conditions were good to determine the consequence of a gene knockdown, especially for investigation of pleiotropic effect on non-direct target proteins. For example, knockdown of the *MKI67* gene with 10 nM concentration affected the expression of several proteins as seen Figure 4. Higher concentrations of the siRNA might have diminished the expression of a greater number of proteins to greater extent. However, such results have a great risk of a wrong conclusion. It needn't be said, but a targeted protein declines following a $(1/2)^n$ and never reaches zero by gene knockdown unless it is highly labile.
- (3) As for growth experiment, we can clearly conclude "Yes or No" at a low risk. There is a wide deviation within results of the control experiment. However, our conclusion has been delivered by standard statistical analysis that compared the experimental groups

with the control. When we stated that two groups were significantly different, P value of the comparison was 0.005, 0.01 or 0.003. These P values were far lower than 0.05, with which one can declare significant difference for comparison of two groups. We feel these results convincing.

- (4) I feel strongly that my work is well suited for this Journal rather than a proteomics specific journal, because I initiated this work with biological interest. Our results just don't fit to proteomics that aims at renovation of subjects, such as, proteomics technology, rate of protein production and degradation, modification of proteins, and deeper insight of a protein-protein interaction. Our work differs from those subjects and has an important meaning in biology.
- (5) Regarding non-specific binding, I am not sure about what Reviewer #1 exactly means by "non-specific binding". However, I can guess two review's meanings: 1) binding of the rabbit ELMSAN1 antibody, and 2) protein-protein interaction of nuclear proteins. As for 1), we have shown, by competitive inhibition with an engineered epitope, that the antibody binds specifically to ELMSAN1 isomers and its partially degraded/post-translationally modified versions. As for 2), we have discussed the nature of the non-specific binding in the discussion section. As we argue, this "non-specific binding" does not appear to be harmful to our conclusion but conversely appears to support organizing the RNAmetasome network to some extent. However, it is important to point out that the RNAmetasome network is formed without involvement of the "non-specific binding" as shown with both cell lines. Thus, to not confuse readers, I have added one sentence, "However, it is important to address that the RNAmetasome network is formed without the "non-specific binding" in the end of the discussion: line 372-373.

Reviewer #2 (Remarks to the Author):

Additional experiments were helpful in strengthening authors' conclusions. However, the manuscript requires proofreading and clarifications, a few examples are provided below:

Line 43: In the phrase "spec of the network" what is "spec"?

This phrase has been revised. It is now "Several proteins of the RNAmetasome network are diminished in Hela.c11, and this diminishment is associated with low expression of MDN1 and elevated MKI67 degradation.": Line 42-43

Line 59: Confusing grammar: "As such, RNA metabolisms are interwound and complicated, but its role can be summarized..."

The phrase has been revised. It is now "As such, RNA metabolic processes are interwound and complicated. However, RNAs involved in these processes can be categorized into two groups based on their function: RNAs for macromolecule biogenesis pathways and RNAs for regulation of genes involved in these pathways.": Line 59-63

Line 76: ELMSNAN1 misspelled

Corrected: Line 78

Line 88: “The experiment was conducted as follows.” This sentence does not connect to the previous description of KDM2A.

Figure of 5b of our previous publication unfortunately does not clearly show presence of proteins whose binding to the bait is inhibited by the CPN domain. This suboptimal feature of the figure occurred when I squeezed X-axis of the figure to fit its size to the page space, as a great number of inhibited proteins are buried in spaces between the bars. However, the squeezed diagram still met my primary aim to show presence of proteins that were stimulated by the CPN domain for their binding to the bait. That is how and why Figure 5c was presented. Now, I find that the figure of the previous publication does not properly include features of the CPN effect, especially the presence of the inhibited proteins.

I agree with the Reviewer #2’s remark. My description of this part was chaotic. Originally, I tried to make this introductory description on ELMSAN1 (including Figure 1a) short as possible. As a result, this part was inappropriate and inadequate. Therefore, this part has now been extensively changed: Supplementary Table 1 has been replaced by new Supplementary Table 1 consisting of values that have been calculated with new equations (shown in Figure 1 legend). Accordingly, Figure 1a has been redrawn, and the text (Line 91-104) has been rewritten. Further, relevant terms have been changed from “silences” to “regulates” (line 106) and from “activate” to “inactivate” (line 107).

Line 91-104: it is now “In that experiment, we prepared nuclear extracts from HEK293T cells expressing EGFP-CPN or EGFP and mixed each extract with the *IGFBPL1* promoter sequence immobilized to magnetic beads. Then, bound proteins were eluted and separated via SDS-PAGE. Finally, slices of the gel were excised and subjected to mass spectrometry analysis (MS). For the current interest as to what proteins are strongly inhibited by CPN, we picked the data from gel slices, 1, 2, 3, and 4 (Figure 5a)¹⁶, calculated the severity of the inhibition, and then plotted the resulting values on an X-Y diagram by combining affinity data of the proteins for the bait (Figure 1a, supplementary Table 1). This diagram enabled us to classify the inhibited proteins into two groups: proteins enriched less than 16-fold and more than that. The former group of proteins were at least partly inhibited non-specifically by CPN as a gray curve drawn in the upper part of the diagram indicates; and therefore, these proteins were not of interest to us. On the other hand, the latter group proteins appeared to be inhibited by a specific interaction between each protein and CPN on the bait; and therefore, these proteins were good candidates for further exploration. Of those, about twenty proteins were severely inhibited by CPN; and we found within this group an interesting protein (Figure 1a, red dot), called ELMSAN1, which has a potential to regulate expression of the *IGFBPL1* gene.”

Line 683: What is "typical" protein?

I meant by “typical” that ELMSAN1 is the best representative of the proteins that are highly enriched with the bait and strongly inhibited by CPN. I have removed “typical” in the new revision.

Line 111-113: This sentence needs rephrasing to clarify the point: “Likewise, epELM-512 inhibited the antibody from recognising most other bands, even though epELM-5 liberated from PVDF and bound to the protein bands and the backgrounds made the inhibition look weak.” clarified. It is now “Likewise, epELM-5 inhibited the antibody’s recognition of other bands. This inhibition looked weaker than expected on a few bands, but this weakness is caused by both

binding of epELM-5 to the bands and absorption of the epitope to the background.”: Line 122-124

Line 126: What is “converse relationship”? Do the authors mean “inverse”?
changed to inverse: Line 134

Line 307: Capitalize “We”
Corrected: Line 316

Reviewer #3 (Remarks to the Author):

In their revised manuscript, Iuchi and Paulo have toned down their claim about the identification of a novel multiprotein “complex” and adequately addressed majority of the concerns that I had after reading the initial version of the paper. Specifically, they verified the existence of the RNA metasome network in another human cell line, namely HeLa, and documented that MDN1 and MKI67 proteins are primarily responsible for the network assembly in different cell lines. Further, they performed additional western-blot analyses focused on interaction between ELMSAN1 and HDAC and histones and modified their claims accordingly to newly obtained results. Specificity of the anti-ELMSAN1 antibodies has been now more reliably proven through additional experiments employing inhibition of protein recognition with ELMSAN1-derived epitope. This makes the coIP-MS data more trustworthy. New set of coIP-MS analysis was also performed to better assess quantitative differences between particular samples. I feel that the results presented in this communication are a good starting point for a comprehensive characterization of the functional importance of the RNA metasome network in a variety of cellular processes. Together with amendments and clarifications in the text, the manuscript by Iuchi and Paulo has been improved and I am now willing to recommend accepting it for publication in Communications Biology.

Minor comments that should be addressed prior to final acceptance:

- extensive editing of English would be desirable, since multiple fragments of the manuscript (particularly in the Results section) are not written clearly enough (e.g. lines 123-127; 180-183); several suggestions for minor text corrections are provided below;

These have been rephrased.

The first phrase is now “# 8 had another peculiarity. It had an inverse relationship with the 140-kDa ELMSAN1 between the HEk293T and the HeLa.c11 (Supplementary Figure 1c), suggesting that #8 is an ELMSAN1 isomer, H7C1L3, that has most of ⁵⁵⁰AA⁶⁰⁰ but not ⁶³¹AA⁶⁵⁷.”: Line 134-137

The second phrase is now “Following these analyses, we performed another duplicate IP-MS experiment (Exp #3 and #4), having two kinds of controls for each experiment: water and nr-IgG in place of the rabbit ELMSAN1 antibody.”: Line 188-191

- what is the exact meaning of the following sentence in the abstract: “However, inherent attenuation 43 of MDN1 and MKI67 diminishes its spec of the network”? Please, rephrase it to make it more understandable. Also, merge the information about existence of the RNAmetasome

network in different human cell lines (the sentence “Hela.c11 also has the RNAmetasome network” must be removed);

This part has been revised to “Several proteins of the RNAmetasome network are diminished in HeLa.c11, and this diminishment is associated with low expression of MDN1 and elevated MKI67 degradation.”: Line 42-44

- The authors must clearly indicate the source of HeLa.c11 cell line (e.g., is it available from ATCC?)

I inherited HeLa.c11 from Dr. Howard Green. He was careful of maintenance cell lines, as the person who established 3T3 immortalization method (Wells, JCB, 168: 988-989, 2005). So, HeLa.c11 kept in his lab must be an excellent cell line. This cell line is not on a list of ATCC and does not seem to be available from anywhere else. Certainly, the culture looks HeLa cell line. Besides above statement, I cannot provide any specific information on HeLa.c11. For this reason, I have just described this cell line as a gift from Howard Green in the method section: Line 403

- line 59: “RNA metabolisms” – change into “RNA metabolic processes”
Changed: Line 60

- lines 60-64 – merge into one sentence, since some information is unnecessarily repeated
Revised, it is now “However, RNAs involved in these processes can be categorized into two groups based on their function: RNAs for macromolecule biogenesis pathways and RNAs for regulation of genes involved in these pathways.” Line 60-63

- line 70: “regulator” – change into “regulatory”
Changed: Line 74

- line 80: remove comma after “and”
Removed: line 82

- line 94: “inhibits ELMSAN1 from binding the IGFBPL1 promoter sequence” – change into inhibits ELMSAN1 binding to the IGFBPL1 promoter sequence”

The phrase had to be improved to describe more detail of the result. The new phrase is as good as your suggestion, I believe.

Line 91-106: It is now “In that experiment, we prepared nuclear extracts from HEK293T cells expressing EGFP-CPN or EGFP and mixed each extract with the *IGFBPL1* promoter sequence immobilized to magnetic beads. Then, bound proteins were eluted and separated via SDS-PAGE. Finally, slices of the gel were excised and subjected to mass spectrometry analysis (MS). For the current interest as to what proteins are strongly inhibited by CPN, we picked the data from gel slices, 1, 2, 3, and 4 (Figure 5a)¹⁶, calculated the severity of the inhibition, and then plotted the resulting values on an X-Y diagram by combining affinity data of the proteins for the bait (Figure 1a, supplementary Table 1). This diagram enabled us to classify the inhibited proteins into two groups: proteins enriched less than 16-fold and more than that. The former group of proteins were at least partly inhibited non-specifically by CPN as a gray curve drawn in the upper part of the diagram indicates; and therefore, these proteins were not of interest to us. On the other hand, the latter group proteins appeared to be inhibited by a specific interaction

between each protein and CPN on the bait; and therefore, these proteins were good candidates for further exploration. Of those, about twenty proteins were severely inhibited by CPN; and we found within this group an interesting protein (Figure 1a, red dot), called ELMSAN1, which has a potential to regulate expression of the *IGFBPL1* gene.”

- lines 105-106: “Therefore, the IP result is indeed valid” – remove this sentence completely or move it after arguments proving that IP results are reliable

The sentence has been removed. Line 116

- line 107 and elsewhere: ”DsiRNAs” – change into “siRNAs”

Changed: Line 117

- line 169: “changes” – convert into “change”

Converted: line 177

- first paragraph of the Discussion should be incorporated into the results section; accordingly, Figure 6 and associated descriptions should be placed earlier in the results section

The paragraph has been moved to the end of the result section: line 315-340

REVIEWERS' COMMENTS:

Reviewer #1 (Remarks to the Author):

The authors have adequately addressed most of my major concerns and overall the manuscript has been substantially improved from the initial submission. Identification of the RNA metasome network is a novel finding and while it would be great to see additional experimental validation of this network the work presented lays a nice foundation for future studies.

Reviewer #2 (Remarks to the Author):

The work by Shiro Iuchi and Joao Paulo contains sufficient experiments to describe the RNAmetasome network by IP-MS. Interpretation of the results of the follow-up studies is complicated by very nuanced changes in protein localization (Figure 4, ctrl vs KD experiments) and interactions observed in IP-WB experiments.

For example, changes in protein localization between KD and ctrl are hard to interpret because they come from different imaging slides and it is not clear whether image acquisition parameters match between the two. The authors claim that MDN1 nucleolar localization is diminished upon MKI67 depletion (Figure 4, p21 vs p17), but these changes are not obvious and very subtle at most to support Fig. 5c mechanism.

It would be helpful if the authors included detailed description of the image data acquisition and how they matched exposure and other parameters between images from different slides. For example, how were the focal plains match? Perhaps I missed this, but quantitation of the imaging data would aid in interpretation of protein co-localization claims in ctrl and KD experiments.

For Western blot data, the authors could repeat these experiments and provide quantitative data based on statistically meaningful number of replicates, which would help strengthen their conclusions. In addition, Fig 5d does not provide support to the mechanism shown in Fig 5b, and, again, the data in image panels Fig4 57-64 is hard to interpret due to very subtle changes in protein staining between KD and control experiments.

In its present form, the manuscript could be a better fit for a proteomics journal, such as Molecular & Cellular Proteomics.

Reviewer #3 (Remarks to the Author):

I am satisfied with further improvements made by the authors in the course of second revision and I recommend publication of the manuscript in its current form in Communications Biology.

REVIEWERS' COMMENTS:

Reviewer #1 (Remarks to the Author):

The authors have adequately addressed most of my major concerns and overall the manuscript has been substantially improved from the initial submission. Identification of the RNA metasome network is a novel finding and while it would be great to see additional experimental validation of this network the work presented lays a nice foundation for future studies.

Thank you for the appreciation and the approval.

Reviewer #2 (Remarks to the Author):

The work by Shiro Iuchi and Joao Paulo contains sufficient experiments to describe the RNAmetasome network by IP-MS. Interpretation of the results of the follow-up studies is complicated by very nuanced changes in protein localization (Figure 4, ctrl vs KD experiments) and interactions observed in IP-WB experiments.

I believe that our interpretation is clear-cut. Please see my reasoning below.

For example, changes in protein localization between KD and ctrl are hard to interpret because they come from different imaging slides and it is not clear whether image acquisition parameters match between the two. The authors claim that MDN1 nucleolar localization is diminished upon MKI67 depletion (Figure 4, p21 vs p17), but these changes are not obvious and very subtle at most to support Fig. 5c mechanism.

I took photos of these images under same conditions that the procedure was described in the Method section. Nevertheless, I do not directly compare intensity of fluorescence between KD and the ctrl samples. For comparison of fluorescent intensity of a protein, such as NIFK and GNL2, its perinuclear fluorescent intensity was used as the internal marker as its intensity was nearly identical between the two groups, perhaps because of stability of these proteins in the compartment. Taking this internal control into account, readers can observe a clear-cut difference between the two groups in the nucleosol. In the case of Figure 4, p21 vs p17, subcellular localisation of MDN1, but not fluorescent intensity of the protein, was compared referring to the DAPI counterstaining. These images show that MDN1 occupied the nucleolus of the ctrl cell while it was not present in the nucleolus of the KD cell. The difference between the two is evident from these data.

It would be helpful if the authors included detailed description of the image data acquisition and how they matched exposure and other parameters between images from different slides. For example, how were the focal plains match? Perhaps I missed this, but quantitation of the imaging data would aid in interpretation of protein co-localization claims in ctrl and KD experiments.

I added further description of the image data acquisition in the methods section, lines 532-536. “Immunofluorescent image was taken by capturing emission derived from 300 msec excitation of both Alexa fluor 555 (red) and Alexa fluor 488 (green) dyes and emission derived from 100 msec excitation of DAPI. Z-plane images of a cell (or a culture field) were taken every 0.25 μm , and these Z-plane images slicing the nucleolus through the centre were used for comparison.”

As for quantification, I didn't do it because I believe that quantification of fluorescent intensity does not adequately reflect the complexity of protein expression in cells.

For Western blot data, the authors could repeat these experiments and provide quantitative data based on statistically meaningful number of replicates, which would help strengthen their conclusions. In addition, Fig 5d does not provide support to the mechanism shown in Fig 5b, and, again, the data in image panels Fig4 57-64 is hard to interpret due to very subtle changes in protein staining between KD and control experiments.

Although we did not repeat experiments sufficiently for statistical analysis, but these Western blot data were reproducible. Therefore, the data we presented provide reliable information.

Fig 5b model was made of Immunostaining data, and it is partially supported by the Fig 5d Western blot data. The reason that the Western blot data does not fully support the model has been provided in the text, Lane 309-312

As for Fig4 57-64, we did not compare intensity of immunofluorescence, but compared subcellular protein localisation between KD and the ctrl. Analogous images are presented in Fig4 65-72, in which the DAPI counter staining shows the MDN1 sublocalisation. These two groups of images strongly support our model. To address consistency of the MDN1 localisation, the description has been slightly changed. It now reads “In these nuclei, a majority of MDN1 was localised apart from MKI67 (Figure 4, P61-64). Likewise, the peculiar localisation of MDN1 was observed in DAPI stained nuclei (P69-72).” Line 300-301

In its present form, the manuscript could be a better fit for a proteomics journal, such as Molecular & Cellular Proteomics.

I am pleased at the suggestion that our work could be a paper for a proteomics journal, such as Molecular & Cellular Proteomics. However, I like to publish our work in this multidisciplinary journal for biology. The reasons are:

1) my interest is in biology; 2) analysis by several different approaches that complement each other to describe phenomena of biology, such as genetics, biochemistry, immunology, cytology, and mass spectrometry are involved; and 3) our conclusion has an important implication in biology.

Reviewer #3 (Remarks to the Author):

I am satisfied with further improvements made by the authors in the course of second revision and I recommend publication of the manuscript in its current form in Communications Biology.

Thank you for the appreciation and the recommendation to Communications Biology.